# Seroconversion stages COVID19 into distinct pathophysiological states

Matthew D Galbraith[1,2], Kohl T Kinning[1], Kelly D Sullivan[1,3], Ryan Baxter[4], Paula Araya[1], Kimberly R Jordan[4], Seth Russell[5], Keith P Smith[1], Ross E Granrath[1], Jessica R Shaw[1], Monika Dzieciatkowska[6], Tusharkanti Ghosh[7], Andrew A Monte[8], Angelo D'Alessandro[6], Kirk C Hansen[6], Tellen D Benett[9], Elena WY Hsieh[4,10], Joaquín M Espinosa[1,2]*

[1]Linda Crnic Institute for Down Syndrome, University of Colorado Anschutz Medical Campus, Aurora, United States; [2]Department of Pharmacology, University of Colorado Anschutz Medical Campus, Aurora, United States; [3]Department of Pediatrics, Division of Developmental Biology, University of Colorado Anschutz Medical Campus, Aurora, United States; [4]Department of Immunology and Microbiology, University of Colorado Anschutz Medical Campus, Aurora, United States; [5]Data Science to Patient Value, University of Colorado Anschutz Medical Campus, Aurora, United States; [6]Department of Biochemistry and Molecular Genetics, University of Colorado Anschutz Medical Campus, Aurora, United States; [7]Department of Biostatistics and Informatics, Colorado School of Public Health, Aurora, United States; [8]Department of Emergency Medicine, University of Colorado Anschutz Medical Campus, Aurora, United States; [9]Department of Pediatrics, Sections of Informatics and Data Science and Critical Care Medicine, University of Colorado Anschutz Medical Campus, Aurora, United States; [10]Department of Pediatrics, Division of Allergy/Immunology, University of Colorado Anschutz Medical Campus, Aurora, United States

*For correspondence:
joaquin.espinosa@cuanschutz.edu

**Abstract** COVID19 is a heterogeneous medical condition involving diverse underlying pathophysiological processes including hyperinflammation, endothelial damage, thrombotic microangiopathy, and end-organ damage. Limited knowledge about the molecular mechanisms driving these processes and lack of staging biomarkers hamper the ability to stratify patients for targeted therapeutics. We report here the results of a cross-sectional multi-omics analysis of hospitalized COVID19 patients revealing that seroconversion status associates with distinct underlying pathophysiological states. Low antibody titers associate with hyperactive T cells and NK cells, high levels of IFN alpha, gamma and lambda ligands, markers of systemic complement activation, and depletion of lymphocytes, neutrophils, and platelets. Upon seroconversion, all of these processes are attenuated, observing instead increases in B cell subsets, emergency hematopoiesis, increased D-dimer, and hypoalbuminemia. We propose that seroconversion status could potentially be used as a biosignature to stratify patients for therapeutic intervention and to inform analysis of clinical trial results in heterogenous patient populations.

## Introduction

COVID19 (coronavirus disease of 2019), the disease caused by the severe acute respiratory syndrome coronavirus 2 (SARS-CoV-2), has caused more than 2.33 million deaths worldwide since late 2019. SARS-CoV-2 is a highly contagious coronavirus that uses angiotensin-converting enzyme-2 (ACE-2), a protein widely expressed on lung type II alveolar cells, endothelial cells, enterocytes, and

arterial smooth muscle cells, as its primary cellular entry receptor (*Hoffmann et al., 2020*). Neuropilin-1 (NRP1) has been characterized as an additional entry receptor for SARS-CoV-2, thus extending the range of host cells and tissues directly affected by the virus (*Cantuti-Castelvetri et al., 2020*; *Daly et al., 2020*). The clinical presentation of COVID19 is highly variable, ranging from asymptomatic infection to multiorgan failure and death (*Wiersinga et al., 2020*). Mild symptoms include a flu-like condition consisting of fever, nasal congestion, cough, fatigue, and myalgia. In a small fraction of patients, SARS-CoV-2 causes more severe effects in multiple organ systems. These include respiratory failure, thromboembolic disease, thrombotic microangiopathies, stroke, neurological symptoms including seizures, as well as kidney and myocardial damage (*Wiersinga et al., 2020*). The molecular and cellular bases of this clinical heterogeneity remain to be elucidated.

Several pathophysiological processes have been implicated in the etiology of severe COVID19 symptoms, including but not restricted to a hyperinflammation-driven pathology (*Tay et al., 2020*), disruption of lung barrier function by Type I and III interferons (IFN) (*Broggi et al., 2020*; *Major et al., 2020*), organ damage by systemic activation of the complement cascade (*Holter et al., 2020*), vascular pathology caused by a bradykinin storm (*Garvin et al., 2020*), and a dysregulated fibrinolytic system (*D'Alessandro et al., 2020*). The interplay between these non-mutually exclusive processes is yet to be fully elucidated, and each of them offers opportunities for therapeutic interventions currently being tested in clinical trials. However, the lack of precise biomarkers for cohort stratification and targeted therapeutics has hampered progress in this area.

We report here the results of a cross-sectional multi-omics analysis of hospitalized COVID19 patients investigating the multidimensional impacts of seroconversion status. When stratifying patients by a quantitative metric of seroconversion, or 'seroconversion index', we were able to define biosignatures differentially associated with humoral immunity. Low seroconversion indices associate with high levels of activated T cells and cytokine-producing natural killer (NK) cells, biosignatures of monocyte activation, high levels of IFN alpha, gamma, and lambda ligands, markers of systemic complement activation, and depletion of lymphocytes, neutrophils, and platelets. In seroconverted patients, all these biosignatures are decreased or fully reversed, leading instead to increased levels of circulating plasmablasts and mature and activated B cell subsets, increased numbers of neutrophils, lymphocytes, and platelets, elevated markers of platelet degranulation and D-dimer, and significantly decreased levels of albumin and major liver-derived proteins, indicative of increased liver damage and/or vascular leakage. Altogether, these results indicate that a quantitative assessment of seroconversion status could be employed to map the trajectory of underlying pathophysiological processes, with potential utility in stratification of patients in the clinic and enhanced interpretation of clinical trial data.

## Results

### Hospitalized COVID19 patients display highly variable seroconversion status

In order to investigate variations in the pathophysiological state of COVID19 patients, we completed an integrated analysis of 105 research participants, including 73 COVID19 patients versus 32 negative controls (*Figure 1a*). Cohort characteristics are summarized in *Supplementary file 1*. COVID19 patients tested positive for SARS-CoV-2 infection by PCR and/or antibody testing and were hospitalized due to COVID19 symptoms, but none of them had developed severe pathology requiring ICU admission at the time of blood collection. The control group was recruited from the same hospital system, where they were receiving care for diverse comorbidities (*Supplementary file 1*) but tested negative for SARS-CoV-2 infection. Research blood draws were obtained from consented participants and analyzed by matched SARS-CoV-2 seroconversion assays, plasma proteomics using two alternative platforms [mass-spectrometry (MS) and SOMAscan assays], 82-plex cytokine profiling using multiplex immunoassays with Meso Scale Discovery (MSD) technology, and immune cell profiling via mass cytometry (MC) (*Figure 1a*) (see Materials and methods).

In order to stratify the COVID19-positive cohort, we measured seroconversion with multiplex immunoassays detecting IgGs against four different SARS-CoV-2 peptides: full length nucleocapsid, full length spike protein (spike), as well as smaller peptides encompassing the N-terminus domain (NTD) and the receptor-binding domain (RBD) of the spike protein (see Materials and methods). The

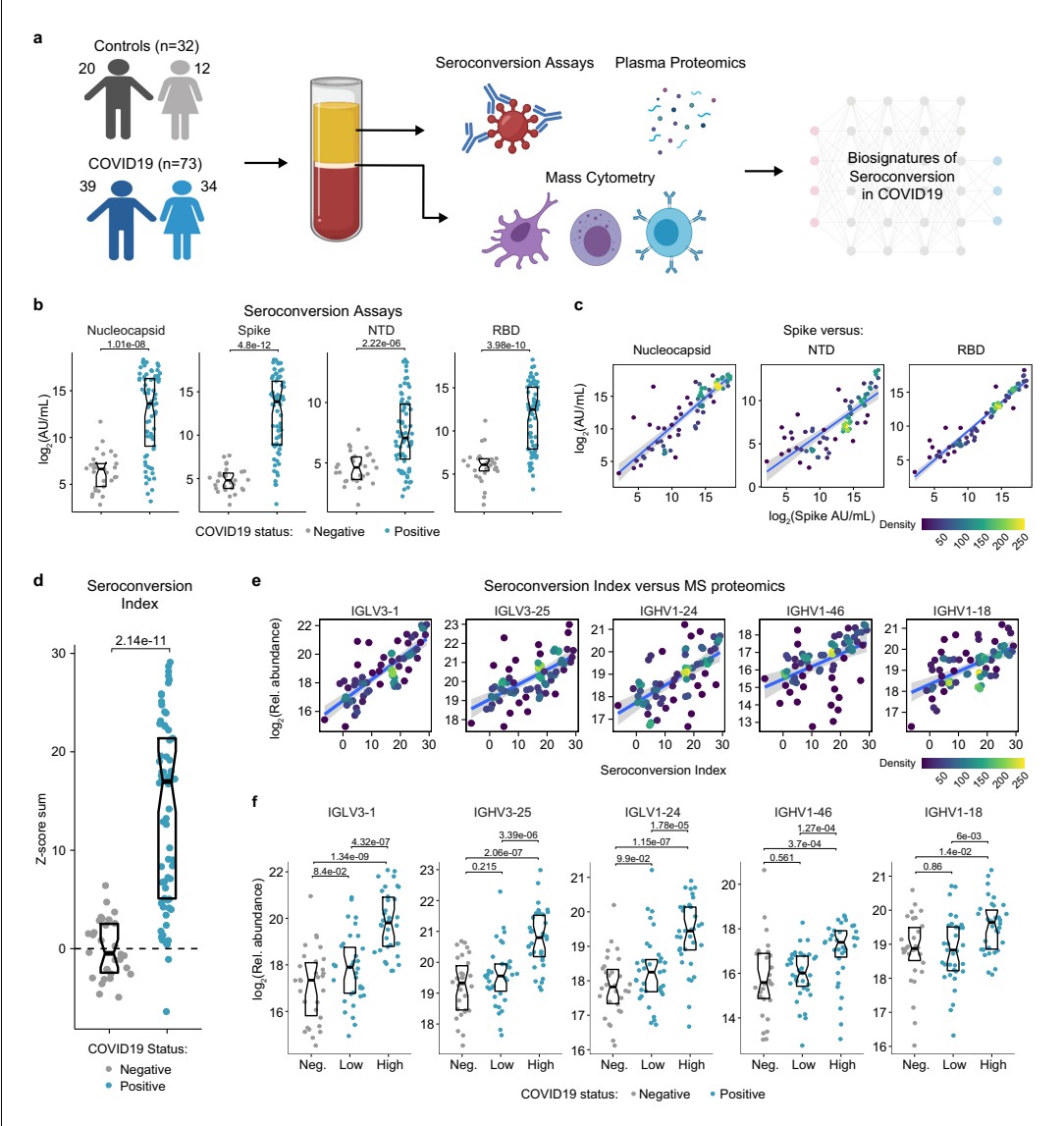

**Figure 1.** Highly variable seroconversion status among hospitalized COVID19 patients. (**a**) Overview of experimental approach. Blood samples from 105 research participants, 73 of them with COVID19, were analyzed by matched multiplex immunoassays for detection of antibodies against SARS-CoV-2, plasma proteomics using mass spectrometry (MS), SOMAscan proteomics, and cytokine profiling using Meso Scale Discovery (MSD) technology. Data was then analyzed to define biosignatures of seroconversion. (**b**) Multiplex immunoassays were used to measure antibodies against the SARS-CoV-2 nucleocapsid and spike proteins, as well as specific peptides encompassing the N-terminus domain (NTD) and receptor-binding domain (RBD) of the spike protein. Data are presented as modified Sina plots with boxes indicating median and interquartile range. Numbers above brackets are p-values for Mann–Whitney tests. (**c**) Scatter plots showing correlations between antibodies against the full-length spike protein versus antibodies against the nucleocapsid, NTD, and RBD domains. Points are colored by density; lines represent linear model fit with 95% confidence interval. (**d**) Seroconversion indices were calculated for each research participant by summing the Z-scores for each of the four seroconversion assays. Z-scores were calculated from the adjusted concentration values for each epitope in each sample, based on the mean and standard deviation of COVID19-negative samples. (**e**) Scatter plots displaying the top five correlations between seroconversion indices and proteins detected in the MS proteomics data set among COVID19 patients. Points are colored by density; lines represent linear model fit with 95% confidence interval. (**f**) Sina plots showing values for the top five proteins correlated with seroconversion comparing the control cohort (Negative, Neg.) to COVID19 patients divided into seroconversion low and high status. Data are presented as modified Sina plots with boxes indicating median and interquartile range. Numbers above brackets are q-values for Mann–Whitney tests. See also *Figure 1—figure supplement 1*.

The online version of this article includes the following figure supplement(s) for figure 1:

**Figure supplement 1.** Biosignatures of seroconversion among hospitalized COVID19 patients.

COVID19 cohort displayed significantly elevated levels of anti-SARS-CoV-2 IgGs in all four assays, with strong inter-individual variability (*Figure 1b*). As a control, levels of antibodies against the Flu A Hong Kong H3 virus strain were no higher in COVID19 patients (*Figure 1—figure supplement 1a*). In COVID19 patients, reactivity against the spike protein correlated positively with reactivity against the other three peptides (*Figure 1c*). Therefore, we generated a seroconversion index by summing Z-scores for each individual seroconversion assay, which enabled us to assign a quantitative seroconversion value to each patient (*Figure 1d*). For the purpose of this study, we divided the COVID19 cohort into equally sized groups of low and high seroconversion indices, referred hereto as sero-low and sero-high groups, respectively.

We then set out to define biosignatures significantly associated, either positively or negatively, with the seroconversion index among COVID19 patients by analyzing correlations with the proteome, cytokine profiling, and MC data sets. When calculating Spearman correlation values between seroconversion indices and individual features in the other data sets, we identified hundreds of proteins and dozens of immune cell types significantly correlated with seroconversion (*Figure 1—figure supplement 1b–g*, *Supplementary files 2–5*). Reassuringly, top positive correlations among 407 abundant plasma proteins detected by MS are dominated by specific immunoglobulin sequences, including several that were previously observed to be enriched in the bloodstream of COVID19 patients during seroconversion (*Supplementary file 2*, *Figure 1e–f*; *Nielsen et al., 2020*).

Altogether, these observations suggest that seroconversion is accompanied by significant changes in underlying pathophysiological processes in COVID19, which prompted us to complete a more thorough analysis of these correlations.

## Immune cell signatures of seroconversion in COVID19

First, we investigated associations between seroconversion and changes in the frequencies of peripheral immune cell subsets among COVID19 patients. Among all live CD45+ white blood cells (WBCs), significant negative associations included plasmacytoid dendritic cells (pDCs), distinct subsets of CD4+ and CD8+ T cells, and CD56$^{bright}$ NK cells (*Figure 2a,b*, *Figure 2—figure supplement 1a,b*). Conversely, positive associations were dominated by B cell subsets. pDCs were only mildly elevated in sero-low COVID19 patients relative to the control group but significantly decreased in the circulation of sero-high patients (*Figure 2c*). Being first responders during a viral infection, pDCs are predicted to be activated and extravasate into the circulation early on as part of the innate immune response, ahead of development of humoral immunity. Their significant reduction in the bloodstream of sero-high patients could be indicative of exhaustion and/or depletion over the course of the disease. Among CD4+ T cells, we observed elevated frequencies of Th1, Th17, Th1/17, follicular helper CD4+ T cells (T$_{FH}$), and terminally differentiated effector memory CD45RA+ subsets in sero-low COVID19 patients, with frequencies falling back to baseline or below baseline in sero-high patients (*Figure 2a,c*). Among CD8+ T cells, a similar behavior was observed for activated (CD95+), effector (T-bet+Eomes+), senescent (T-bet+ Eomes−), effector memory, and terminally differentiated CD45RA+ subsets (*Figure 2a–c*, *Figure 2—figure supplement 1a,b*). These patterns were largely conserved when calculating frequencies within all T cells and within CD4+ and CD8+ T cell subsets (*Figure 2—figure supplement 1a*, *Supplementary file 5*). These changes in peripheral T cell subsets are consistent with an acute and transient antiviral T cell response in patients with low seroconversion indices, marked by elevated levels of activated and effector CD8+ T cells, polarization of CD4+ T cells toward the Th1 antiviral state, accompanied by development of T cell memory, T$_{FH}$-assisted maturation of B cells, and eventual senescence and terminal differentiation of cytotoxic CD8+ T cells. Notably, we also observed increases in CD3+ CD4- CD8- T cells (DN T cells) only in sero-low patients (*Figure 2a*). DN T cells display distinct effector phenotypes, including an upregulated cytolytic machinery, and may mediate tissue damage in autoinflammatory conditions such as systemic lupus erythematosus and Sjogren's syndrome (*Brandt and Hedrich, 2018*). This bimodal T cell behavior is accompanied by increases in the frequency of CD56$^{bright}$ NK cells only in sero-low patients (*Figure 2a,c*). CD56$^{bright}$ NK cells lack expression of inhibitory receptors and express high levels of activating receptors, cytokine and chemokine receptors, and adhesion molecules (*Poli et al., 2009*). Although CD56$^{bright}$ NK cells are not as cytotoxic as other NK subsets, they are strong producers of key cytokines involved in the immune response, most prominently IFNG, which we found to be elevated in sero-low patients (see later, *Figure 3*). CD56$^{bright}$ NK cells have been found to be elevated in some autoimmune conditions and infections (*Poli et al., 2009*).

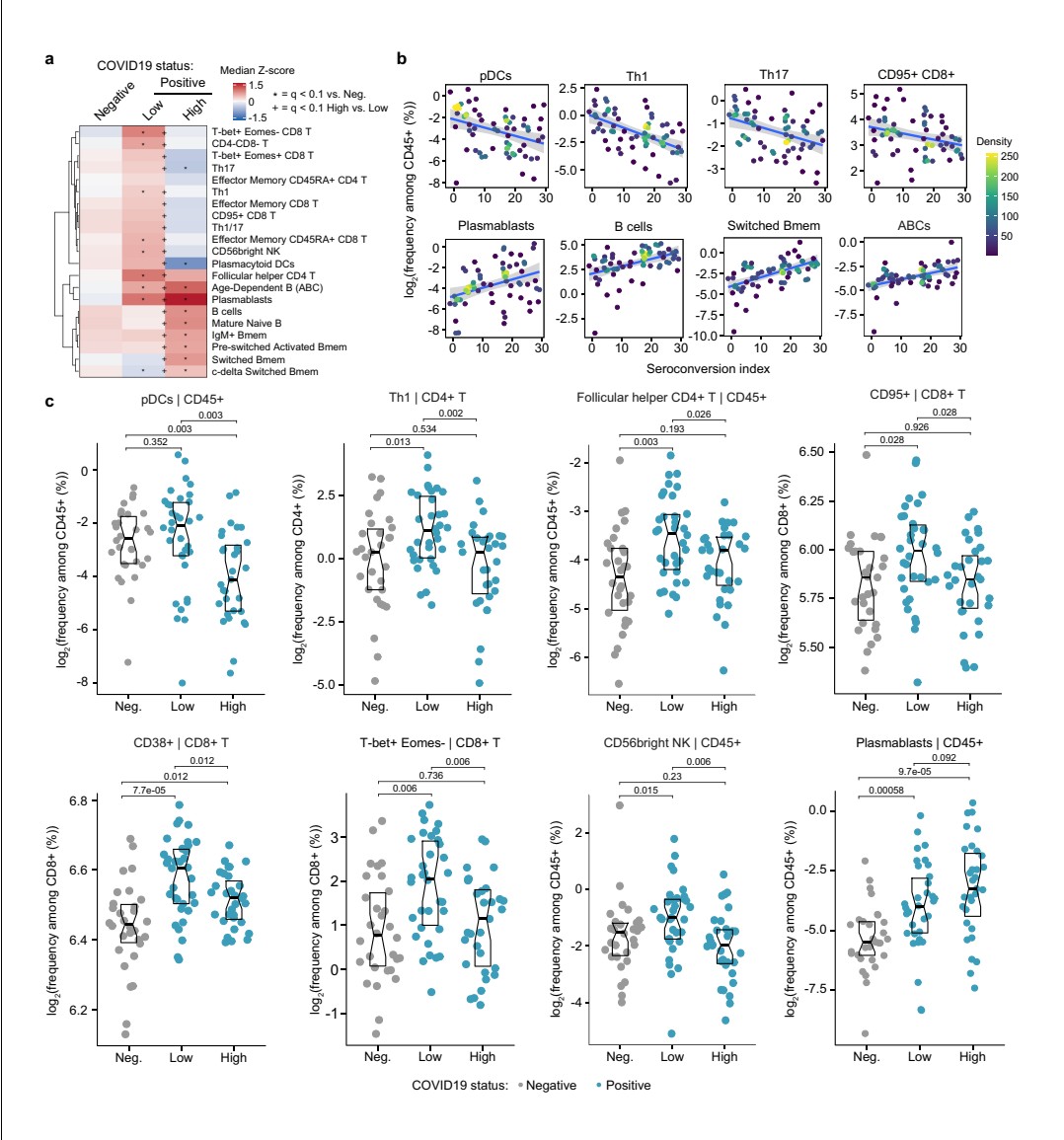

**Figure 2.** Seroconversion associates with significant changes in peripheral immune cell frequencies. (a) Heatmap representing changes in the frequency of immune cell subsets that are significantly correlated, either positively or negatively with seroconversion status. Values displayed are median Z-scores, derived from cell frequencies among all CD45+ cells, for each cell subset for controls (negative, Neg.) versus COVID19 patients divided into seroconversion low (Low) and high (High) status. Z-scores were calculated from the adjusted frequency values for each cell type in each sample, based on the mean and standard deviation of COVID19-negative samples. Asterisks indicate a significant difference relative to the control COVID19-negative group, and the + symbols indicate a significant difference between sero-low and sero-high groups after multiple hypothesis correction (q < 0.1, Mann–Whitney test). (b) Scatter plots for indicated immune cell types significantly correlated with seroconversion indices among COVID19 patients. Points are colored by density; lines represent linear model fit with 95% confidence interval. (c) Sina plots showing values for indicated immune cell types significantly correlated with seroconversion indices among COVID19 patients. The parent cell lineage is indicated in the header and Y axis label for each plot. Data are presented as modified Sina plots with boxes indicating median and interquartile range. Numbers above brackets are q-values for Mann–Whitney tests. See also *Figure 2—figure supplement 1*.

The online version of this article includes the following figure supplement(s) for figure 2:

**Figure supplement 1.** Immune cell signatures of seroconversion.

Somewhat expectedly, the frequency of total B cells and plasmablasts among CD45+ live cells increased with seroconversion (*Figure 2a–c*, *Figure 2—figure supplement 1a–c*). Other B cell subsets displaying significant positive association with seroconversion include key memory subsets such as switched Memory B cells (Switched Bmem), IgM+ memory B cells (IgM+ Bmem), c-delta switched

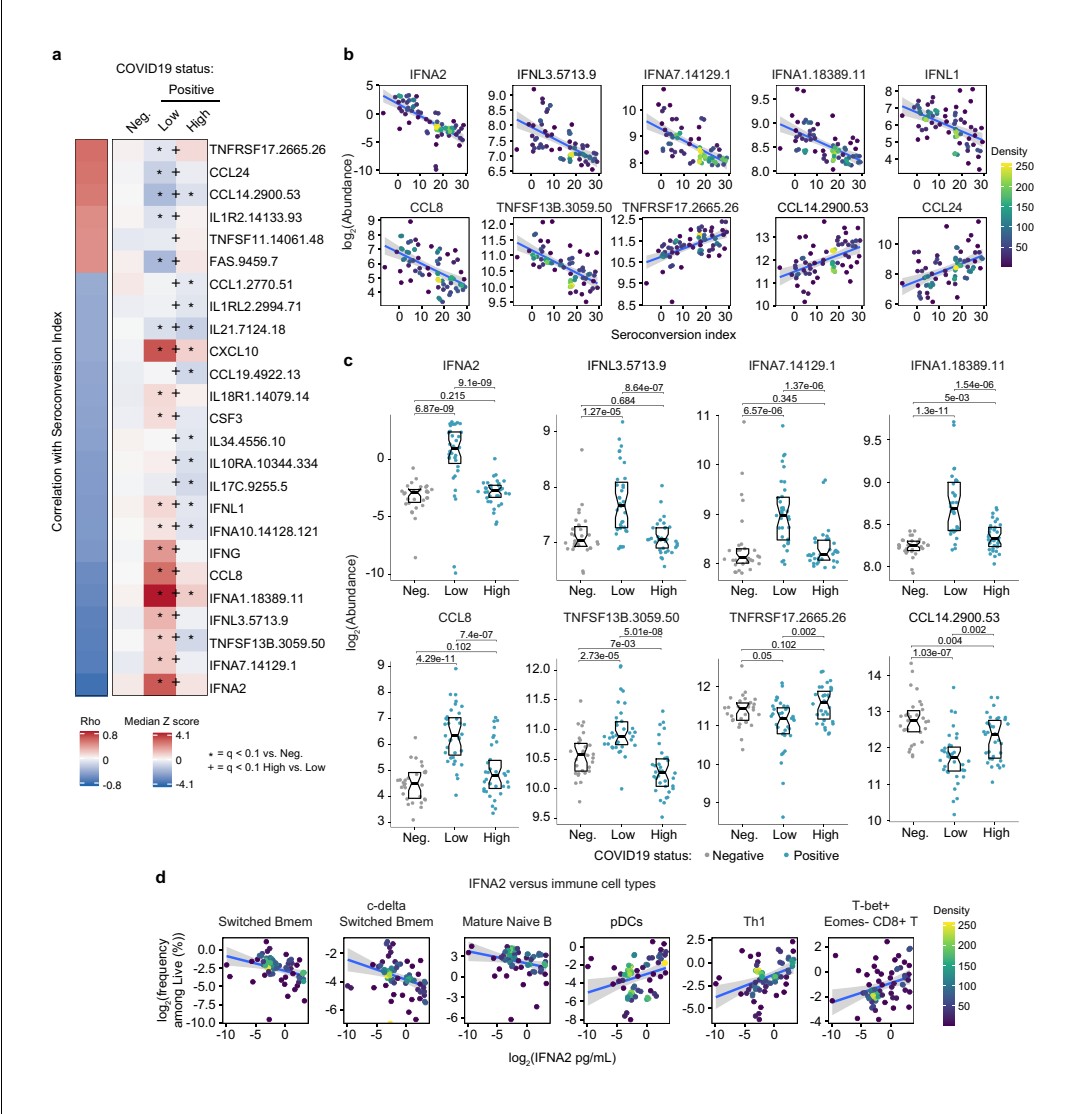

**Figure 3.** Seroconversion is associated with decreased interferon signaling. (**a**) Heatmap displaying changes in circulating levels of immune factors that are significantly correlated, either positively or negatively, with seroconversion status. The left column represents Spearman *rho* values, while the right columns display median Z-scores for each immune factor for controls (negative, Neg.) versus COVID19 patients divided into seroconversion low (Low) and high (High) status. Factors are ranked from most positively correlated (top, high *rho* values) to most anti-correlated (bottom, low *rho* values) with seroconversion index. Z-scores were calculated from the adjusted concentration values for each immune factor in each sample, based on the mean and standard deviation of COVID19-negative samples. Asterisks indicate a significant difference relative to the control COVID19-negative group, and the + symbols indicate a significant difference between sero-low and sero-high groups (q < 0.1, Mann–Whitney test). (**b**) Scatter plots for indicated immune factors significantly correlated with seroconversion indices among COVID19 patients. Points are colored by density; lines represent linear model fit with 95% confidence interval. (**c**) Sina plots showing values for immune factors correlated with seroconversion comparing controls (Neg.) to COVID19 patients divided into seroconversion low and high status. Data are presented as modified Sina plots with boxes indicating median and interquartile range. Numbers above brackets are q-values for Mann–Whitney tests. (**d**) Scatter plots showing correlations between circulating levels of IFNA2 measured by MSD and the indicated cell types measured by mass cytometry. Values for immune cells correspond to frequency among all live cells. Points are colored by density; lines represent linear model fit with 95% confidence interval. See also *Figure 3—figure supplement 1*.

The online version of this article includes the following figure supplement(s) for figure 3:

**Figure supplement 1.** Seroconversion associates with differential abundance of circulating immune factors.

memory B cells (c-delta switched B-mem), and pre-switched activated memory B cells (pre-switched activated Bmem) (*Figure 2a–c*, *Figure 2—figure supplement 1a–c*). Other B cell subsets enriched in sero-high COVID19 patients include mature naïve B cells and age-dependent B cells (ABCs) (*Figure 2a,b*, *Figure 2—figure supplement 1a–c*). An increase in mature naïve B cells is consistent

with the development of humoral immunity. ABCs are associated with typical aging and development of autoimmunity, but their potential role during viral infections is less understood (**Karnell et al., 2017**).

Altogether, these findings illustrate the heterogenous immune state among hospitalized COVID19 patients, with seroconversion status being clearly associated with specific changes in circulating immune cell subsets, which could be largely understood as part of the progression of the antiviral immune response from innate cellular immunity to adaptive humoral immunity. As discussed later, these changes in immune cell frequencies occur in the context of clear depletion of total lymphocytes, neutrophils, and platelets in sero-low patients, with recovery of all these blood cell types in sero-high patients (see later, Figures 5 and 6).

## Seroconversion associates with decreased IFN signaling

Next, we investigated associations between seroconversion and circulating levels of cytokines, chemokines, and other immune modulators in the bloodstream. Toward this end, we analyzed significant correlations in the MS proteomics, SOMAscan proteomics, and MSD data sets (**Figure 1—figure supplement 1b**, **Supplementary files 2–4**). Collectively, these three data sets contain data on dozens of factors involved in immune control (**Supplementary file 6**). The most obvious result from this analysis was a clear negative correlation between seroconversion and circulating levels of key IFN ligands. Among 82 immune factors in the MSD data set, top negative correlations are IFNA2, IFNL1, and IFNG (**Figure 1—figure supplement 1b**, **Figure 3a,b**, **Supplementary files 4** and **6**). Among 5000+ epitopes measured by SOMAscan, IFNA7, IFNL3, and IFNA1 rank among the top 10 negative correlations with seroconversion (**Figure 3a,b**, **Supplementary file 3**). All these IFN ligands were significantly higher in sero-low COVID19 patients relative to the control cohort, but levels fall back within normal ranges in sero-high COVID19 patients (**Figure 3a,c**). These results could be interpreted as a transient wave of IFN production during early stages of SARS-CoV-2 infection, with return to normal levels upon development of humoral immunity. This notion is further supported by the elevated plasma levels of key IFN-inducible proteins, such as CXCL10 (C-X-C Motif Chemokine Ligand 10, IFN-inducible protein 10, IP10), and elevated expression of IFN-inducible mRNAs (e.g. CXCL10, ISG15, MX1, and IFIT1), preferentially in sero-low patients (**Figure 3a**, **Figure 3—figure supplement 1a–c**). Notably, this pattern was not evident for IFNB1 (**Figure 3—figure supplement 1a,b**). Factors involved in monocyte differentiation and activation were also preferentially elevated in sero-low patients, such as CCL8 (C-C Motif Chemokine Ligand 8, Monocyte Chemoattractant Protein 2, MCP2), CSF3 (Colony Stimulating Factor 3, Granulocyte Colony Stimulating Factor, G-CSF), and CCL19 (C-C Motif Chemokine Ligand 19, Macrophage Inflammatory Protein 3 beta, MIP3beta) (**Figure 3a–c**). Although circulating levels of total monocytes and monocyte subsets are not significantly correlated with seroconversion status (**Supplementary file 5**), these results are consistent with a transient round of activation and mobilization of tissue-resident monocytes and macrophages by local IFN production, with subsequent decreases upon seroconversion. In support of this notion, we noticed that circulating levels of CD14, a surface marker for monocytes and macrophages, were strongly anticorrelated with seroconversion, being significantly elevated among sero-low patients and significantly depleted in sero-high patients (**Figure 3—figure supplement 1a,b**). In fact, CD14 was the top negative correlation in the MS proteomics dataset (**Figure 1—figure supplement 1b**, **Supplementary file 2**).

In agreement with the signs of B cell maturation and differentiation associated with seroconversion (**Figure 2a,b** and **Figure 2—figure supplement 1a–c**), top correlations among immune factors include TNFSF13B (TNF Superfamily Member 13B, B-cell activating factor, BAFF), and its receptor, TNFRSF17 (TNF Receptor Superfamily Member 17, B cell Maturation Protein, BCMA). TNFSF13B is increased preferentially in sero-low patients relative to the control group (**Figure 3c**). In contrast, its receptor TNFRSF17 decreases preferentially in sero-low patients, returning to levels similar to the control group upon seroconversion (**Figure 3c**). The increased levels of TNFSF13B in sero-low patients are consistent with a strong wave of B cell stimulation and proliferation prior to B cell maturation and seroconversion. The decrease in circulating soluble TNFRSF17 could be interpreted as a consequence of transient lymphopenia prior to seroconversion (see later, Figure 6). Other interesting top correlations reveal that seroconversion associates with a restoration of circulating cytokines depleted preferentially in sero-low COVID19 patients, such as CCL14 and CCL24 (Eotaxin-2) (**Figure 3a**). Again, these changes could be explained by decreases in lymphocyte counts

preferentially in sero-low patients (see later, Figure 6). Of note, seroconversion is not strongly correlated with changes in acute phase proteins that are commonly elevated upon viral and bacterial infections, such as C-reactive protein (CRP) and ferritin (FTL) (*Supplementary files 2–4*). Whereas CRP levels measured by MS decrease in sero-high patients, ferritin levels remain high (*Figure 3—figure supplement 1d*), suggesting that seroconversion attenuates but does not fully reverse the broader inflammatory phenotype of COVID19.

In order to understand how these changes in cytokines could be integrated with changes observed in circulating immune cell types in the MC data set, we interrogated whether levels of IFNA2, the top anticorrelated cytokine with seroconversion indices, showed significant correlations with immune cell subsets among all live peripheral blood mononuclear cells (PBMCs) (*Supplementary file 7*). Indeed, IFNA2 levels correlated negatively with key B cell subsets increased upon seroconversion, and positively with pDCs, T cell subsets decreased upon seroconversion, and CD56$^{bright}$ NK cells (*Figure 3d*, *Figure 3—figure supplement 1e*, *Supplementary file 7*).

Altogether, these observations could be interpreted as an orchestrated movement in the immune system away from an innate immune response marked by IFN production and IFN-inducible changes in immune cell type frequency and function, toward a state of adaptive humoral immunity and antibody production.

## Seroconversion associates with decreased markers of systemic complement activation

Analysis of the top negative correlations with the MS and SOMAscan proteomics data sets revealed that seroconversion correlates strongly with decreased plasma levels of subunits of the various complement pathways (*Supplementary files 2,3*). In fact, 10 of the top 20 negative correlations in the MS data set are complement subunits or complement regulators, and the top negative correlation in the SOMAscan data set is the complement subunit C1QC (*Figure 4a,b*, *Figure 1—figure supplement 1b,c*, *Supplementary files 2,3*). This led us to complete a more thorough investigation of the interplay between seroconversion and the complement pathways (*Supplementary file 8*).

There are three recognized complement pathways, known as the classical, lectin, and alternative pathways, with significant crosstalk among them and convergence on the so-called terminal pathway that leads to formation of the membrane attack complex (MAC) (*Noris and Remuzzi, 2013*). Proteins from all three pathways were significantly anti-correlated with seroconversion including C1QA, C1QB, C1QC, C1R, and C1S, all involved in initiation of the classical pathway; C2, C4A, and C4B, which share functions in activation of the classical and lectin pathways; C3, which acts both in the lectin and alternative pathways; as well as C6, C7, C8A, C8B, C8G, and C9, which act in the downstream terminal pathway (*Figure 4a*). Additionally, seroconversion correlates negatively with positive regulators of the complement cascade, such as GC (GC Vitamin D Binding Protein), which enhances the chemotactic activity of C5 alpha for neutrophils in inflammation and mediates macrophage activation (*Kew et al., 1995*); FCN3 (Ficolin 3), a protein involved in activation of the lectin complement pathway (*Hein et al., 2010*); CFB (Complement Factor B, C3/C5 Convertase); and CFP (Complement Factor P, Properdin). Most of these factors are significantly elevated in sero-low COVID19 patients relative to controls, but return to baseline or below baseline levels in sero-high patients (*Figure 4c* and *Figure 4—figure supplement 1a,b*). Negative modulators of complement function showed similar behaviors, such as CFH (Complement Factor H), C4BPB (C4b binding protein), and SERPING1 (C1 inhibitor), suggesting the induction of negative feedback mechanisms during complement activation in sero-low patients (*Figure 4a*). Only SOMAscan signals for C5 and the C5.C6 complex showed the opposite behavior, with lower signals in sero-low COVID19 patients, which could be interpreted as increased consumption of the C5 precursor polypeptide by the C5 convertase (*Figure 4a*, *Figure 4—figure supplement 1a,b*).

Altogether, although our proteomics platforms do not enable a complete characterization of the complement cascade in terms of measuring cleaved fragments, protein complexes, and post-translational modifications, our results can nonetheless be understood as a profound, yet transient wave of systemic activation of the complement cascades early during the course of SARS-CoV-2 infections, followed by return to normal levels upon seroconversion.

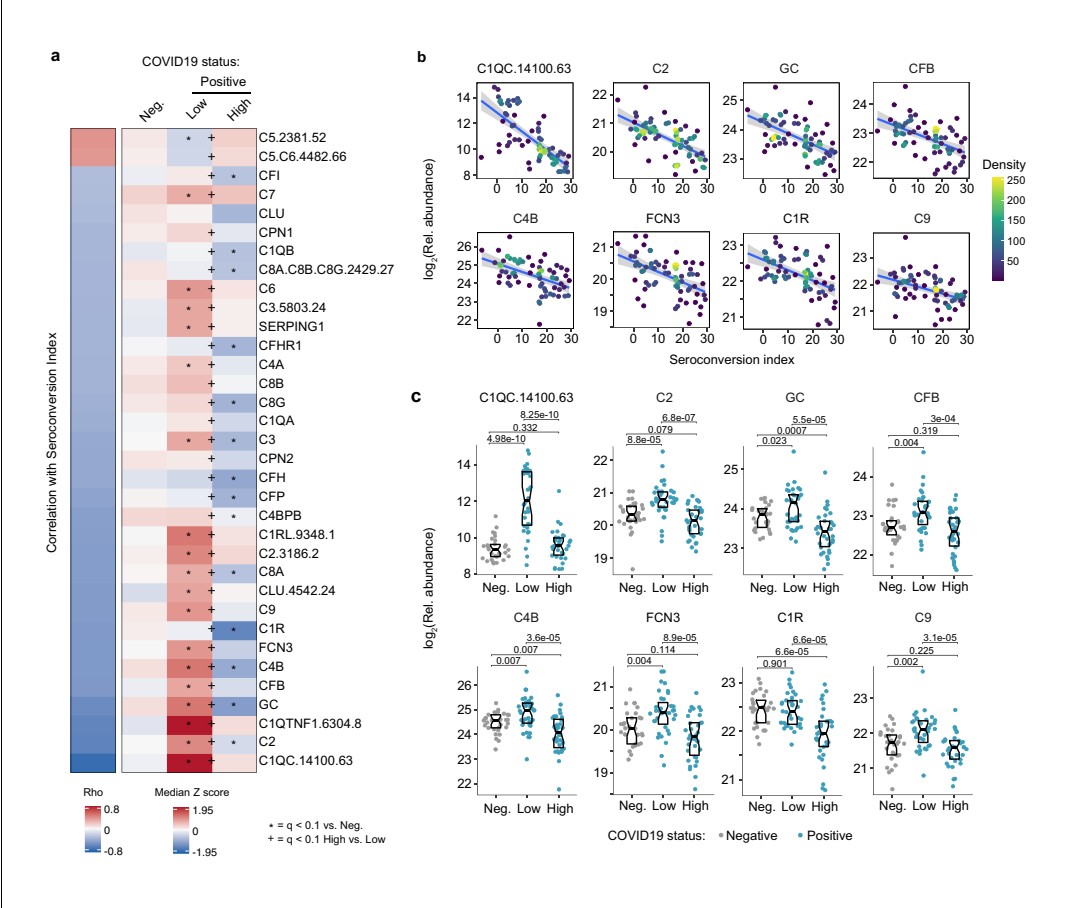

**Figure 4.** Seroconversion correlates with decreased markers of systemic complement activation. (a) Heatmap displaying changes in circulating levels of components of the various complement pathways that are significantly correlated, either positively or negatively, with seroconversion status. The left column represents Spearman *rho* values, while the right columns display median Z-scores for each complement factor for controls (negative, Neg.) versus COVID19 patients (positive) divided into seroconversion low (Low) and high (High) status. Factors are ranked from most positively correlated (top, high *rho* values) to most anti-correlated (bottom, low *rho* values) with seroconversion status. Z-scores were calculated from the adjusted concentration values for each analyte in each sample, based on the mean and standard deviation of COVID19-negative samples. Asterisks indicate a significant difference relative to the control COVID19-negative group, and the + symbols indicate a significant difference between sero-low and sero-high groups (q < 0.1, Mann–Whitney test). (b) Scatter plots for indicated complement factors significantly correlated with seroconversion indices among COVID19 patients. Points are colored by density; lines represent linear model fit with 95% confidence interval. (c) Sina plots showing values for complement factors correlated with seroconversion comparing controls (Negative, Neg.) to COVID19 patients divided into seroconversion low (Low) and high (high) status. Data are presented as modified Sina plots with boxes indicating median and interquartile range. Numbers above brackets are q-values for Mann–Whitney tests. See also *Figure 4—figure supplement 1*.

The online version of this article includes the following figure supplement(s) for figure 4:

**Figure supplement 1.** Seroconversion associates with decreased markers of systemic complement activation.

## Seroconversion associates with remodeling of the hemostasis network toward platelet recovery and activation

Top positive and negative correlations between seroconversion indices and the proteomics data sets included many prominent regulators of hemostasis (*Supplementary files 2,3*). Given the importance of thromboembolism and microangiopathies in COVID19, we decided to investigate the interplay between seroconversion and circulating levels of factors involved in coagulation and thrombosis in more detail (*Supplementary file 9*). Among these factors, positive correlations with seroconversion indices were dominated by markers of platelet degranulation (*Figure 5a*), including key proteins stored in platelet alpha granules, such as SERPINA3 (alpha-1-antichymotrypsin, ACT), PDGFD (platelet derived growth factor D), SELP (selectin P), and GP1BA (glycoprotein Ib alpha) (*Figure 5a–c*, *Supplementary file 9*). Except for SERPINA3, all factors are depleted in sero-low patients relative to

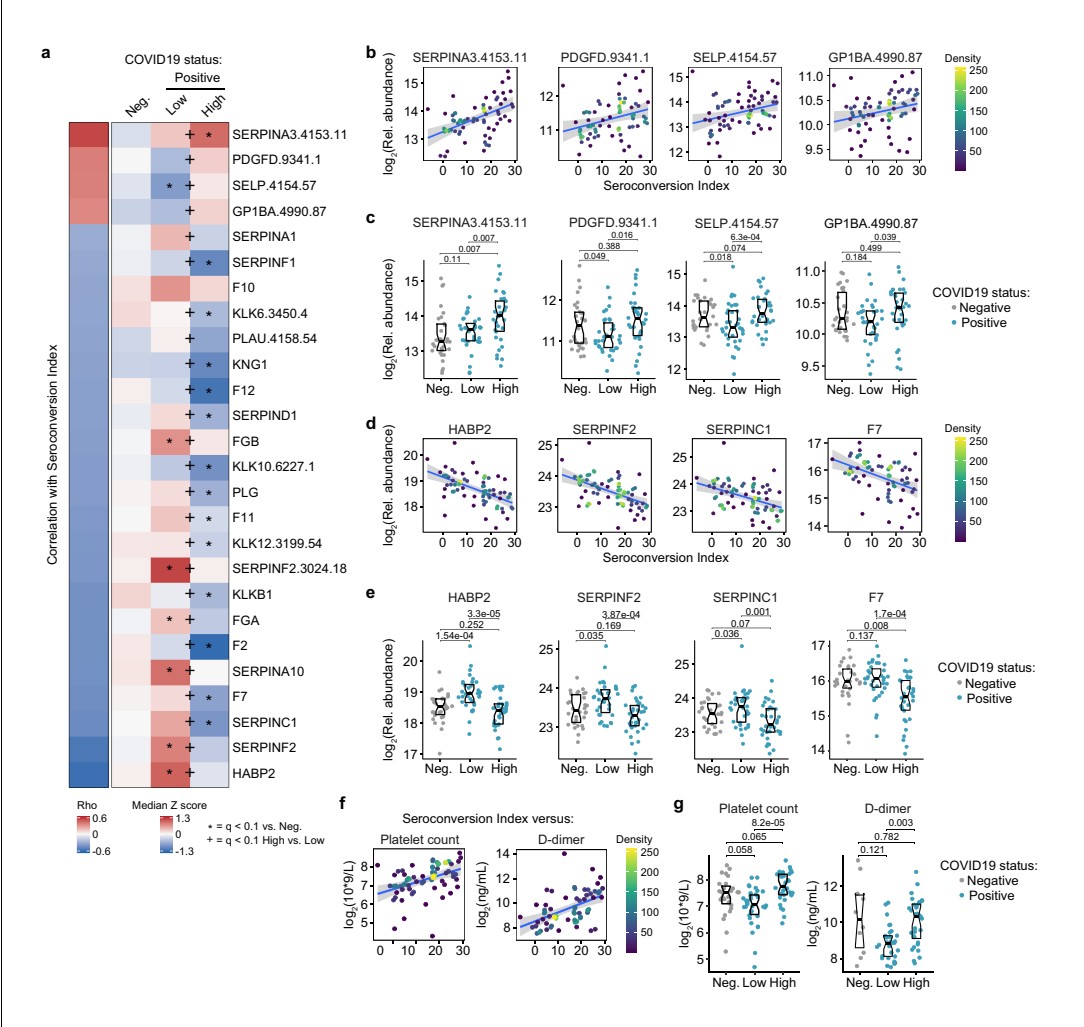

**Figure 5.** Seroconversion associates with remodeling of the hemostasis control network. (a) Heatmap displaying changes in circulating levels of known modulators of hemostasis that are significantly correlated, either positively or negatively, with seroconversion status. The left column represents Spearman *rho* values, while the right columns display row-wise Z-scores for each factor for controls (negative, Neg.) versus COVID19 patients divided into seroconversion low (Low) and high (High) status. Factors are ranked from most positively correlated (top, high *rho* values) to most anti-correlated (bottom, low *rho* values) with seroconversion status. Z-scores were calculated from the adjusted concentration values for each analyte in each sample, based on the mean and standard deviation of COVID19-negative samples. Asterisks indicate a significant difference relative to the control COVID19-negative group, and the + symbols indicate a significant difference between sero-low and sero-high groups (q < 0.1, Mann–Whitney test). (b and c) Scatter plots (b) and Sina plots (c) for factors positively correlated with seroconversion indices. (d and e) Scatter plots (d) and Sina plots (e) for factors negatively correlated with seroconversion indices. (f and g) Scatter plot (f) and Sina plots (g) displaying the correlations between seroconversion index and platelet counts and D-dimer values obtained from clinical laboratory testing. Points in (b), (d), and (f) are colored by density; lines represent linear model fit with 95% confidence interval. Data in (c), (e), and (g) are presented as modified Sina plots with boxes indicating median and interquartile range. Numbers above brackets are q-values for Mann–Whitney tests. See also *Figure 5—figure supplement 1*.

The online version of this article includes the following figure supplement(s) for figure 5:

**Figure supplement 1.** Seroconversion associates with remodeling of the hemostasis network.

the control cohort, and all four factors are significantly elevated in sero-high patients relative to sero-low patients (*Figure 5a,c*).

Conversely, top negative correlations with seroconversion in both proteomics data sets include key modulators of the intrinsic and extrinsic coagulation pathways. Many positive regulators of coagulation are elevated in sero-low patients and depleted in sero-high patients, including HABP2 (hyaluronan activated binding protein 2, factor VII activating protein, FSAP), SERPINF2 (alpha-2-plasmin

inhibitor), F7 (Coagulation Factor VII), F2 (Coagulation Factor II, thrombin), F11 (Coagulation Factor XI), F12 (Coagulation Factor XII), and F10 (Coagulation Factor X), as well as the structural components FGA (Fibrinogen Alpha Chain) and FGB (Fibrinogen Beta Chain) (*Figure 5a,d,e*, *Figure 5—figure supplement 1a,b*). However, many endogenous anticoagulants and drivers of fibrinolysis also show a similar pattern, such as SERPINC1 (antithrombin, AT-III), SERPINA10 (antitrypsin), PLG (Plasminogen), and PLAU (Plasminogen Activator, Urokinase). SERPINC1 is a potent inhibitor of thrombin, as well as coagulation factors IXa, Xa, and XIa (*Rau et al., 2007*). SERPIN10A is another inhibitor of coagulation factors Xa and XIa (*Rau et al., 2007*). Plasminogen is the precursor of plasmin, the key enzyme in fibrinolysis, and PLAU mediates proteolytic generation of plasmin. Lastly, several kallikreins, including KLKB1, KLK12, KLK10, and KLK12, as well as KNG1 (kininogen) are all depleted in seroconverted patients (*Figure 5—figure supplement 1a,b*). The kinin-kallikrein system plays key roles in coagulation, inflammation, and blood pressure control (*Wu, 2015*). Once activated, kallikreins function as serine proteases that can cleave plasminogen into plasmin, thus promoting fibrinolysis, but also high-molecular weight kininogen (HMWK) into the vasoactive peptide bradykinin, thus promoting vasodilation (*Wu, 2015*).

Overall, the interpretation of varying plasma levels for these various modulators of hemostasis is not straightforward, as many of these factors are subject to proteolytic cleavage, consumption, and/or aggregation, and many of them are produced by the liver and cleared by the kidney, two organs affected in COVID19. Therefore, in order to place these complex proteomic signatures in the context of COVID19 pathology, we investigated correlations between seroconversion indices and clinical laboratory values for platelets and D-dimer measured in the course of hospitalization. As part of standard of care in hospitalized COVID19 patients, platelets are routinely counted to assess thrombocytopenia, whereas D-dimer, a proteolytic product of blood clots during fibrinolysis, is routinely measured to assess thrombotic risk. We therefore obtained platelet counts and D-dimer values from the clinical laboratory tests closest in time to the research blood draw employed for the -omics measurements. Both platelet counts and D-dimer correlated positively with the seroconversion scores (*Figure 5f*). Platelet counts were significantly lower in the sero-low group relative to the sero-high group, with ~30% of sero-low patients being considered thrombocytopenic (*Figure 5g*, *Supplementary file 1*).

D-dimer levels were significantly higher in the sero-high group relative to the sero-low group, with the mean value for this group being well above the accepted threshold of 500 ng/mL (*Supplementary file 1*). Very few measurements of D-dimer were available for the control group, which were highly variable and not significantly different from the sero-low group (*Figure 5g*). Altogether, these results are consistent with transient platelet depletion early in the course of SARS-CoV-2 infections, followed by recovery in platelet counts, increased platelet degranulation, and higher levels of fibrinolysis products in sero-high patients. Additionally, as discussed next, depletion of coagulation factors produced by the liver in sero-high patients could be tied to liver dysfunction and/or vascular leakage.

## Seroconversion is accompanied by emergency hematopoiesis and hypoalbuminemia

Analysis of the clinical laboratory values obtained closest in time to the research blood draws revealed other significant correlations between key clinical parameters and seroconversion indices. In addition to the aforementioned positive correlations with platelet counts and D-dimer levels, seroconversion correlated positively and significantly with absolute neutrophil count (ANC), WBC count, and absolute lymphocyte count (ALC) (*Figure 6a–c*, *Figure 6—figure supplement 1a*, *Supplementary file 10*). Consistently, these parameters are lower in the sero-low COVID19 patients, with mean values toward the low end of normal ranges (*Supplementary file 1*), but increase significantly in sero-high patients. These changes are indicative of an early but transient depletion of lymphocytes, neutrophils, and platelets in COVID19, followed by 'emergency hematopoiesis', a compensatory phenomenon involving broad stimulation of hematopoietic and stem cell progenitors (HSPCs) by factors such as G-CSF (CSF3) (*Fuchs et al., 2019*), which we found to be elevated in sero-low patients (*Figure 3a*). The oscillations in neutrophils and lymphocyte counts could explain some of the changes in soluble immune factors depleted in sero-low patients only (e.g. TNFRSF17 expressed by B cells) (*Figure 3a–c*).

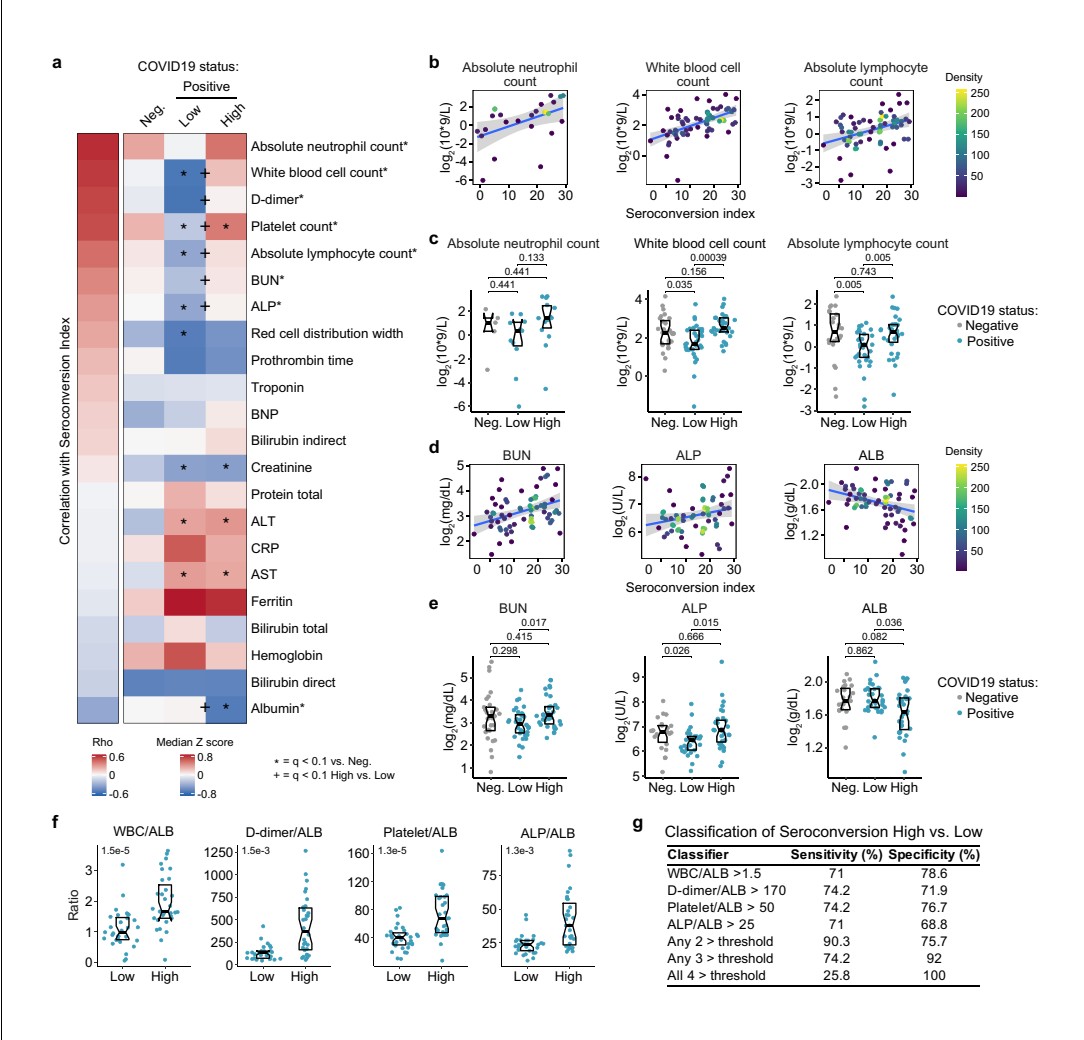

**Figure 6.** Seroconversion associates with recovery in blood cell numbers and hypoalbuminemia. (a) Heatmap displaying correlations between clinical laboratory values and seroconversion status. The left column represents Spearman *rho* values, while the right columns display row-wise Z-scores for each variable for controls (negative, Neg.) versus COVID19 patients divided into seroconversion low (Low) and high (High) status. Measures are ranked from most positively correlated (top, high *rho* values) to most anti-correlated (bottom, low *rho* values) with seroconversion status. Asterisks after the clinical parameter name indicate a significant correlation. Z-scores were calculated from the adjusted concentration values for each analyte in each sample, based on the mean and standard deviation of COVID19-negative samples. Asterisks indicate a significant difference relative to the control COVID19-negative group, and the + symbols indicate a significant difference between sero-low and sero-high groups (q < 0.1, Mann–Whitney test). (b–e) Scatter plots (b and c) and Sina plots (c and e) for indicated clinical laboratory values significantly correlated with seroconversion indices among COVID19 patients. In b and c, points are colored by density; lines represent linear model fit with 95% confidence interval. In c and e, Sina plots show values for clinical laboratory tests correlated with seroconversion comparing controls (Negative, Neg.) to COVID19 patients divided into seroconversion low (Low) and high (High) status. Data are presented as modified Sina plots with boxes indicating median and interquartile range. Numbers above brackets are q values for Mann–Whitney tests. (f) Differences in the indicated ratios of clinical laboratory values between sero-low and sero-high COVID19 patients. Data are presented as modified Sina plots with boxes indicating median and interquartile range. Numbers at upper left of each plot are p-values for Mann–Whitney tests. (g) Table showing how the indicated ratios of the specified clinical values could be potentially used to gauge the seroconversion status of a hospitalized patient with moderate pathology. The units employed for calculating these ratios are $10^3$/mcL for white blood cells (WBC) and platelets; g/dL for albumin (ALB); ng/mL for D-dimer; and U/L for alkaline phosphatase (ALP). ALT: alanine aminotransferase, AST: aspartate aminotransferase; BUN: blood urea nitrogen; BNP: brain natriuretic peptide; CRP: C-reactive protein. See also *Figure 6—figure supplement 1*.

The online version of this article includes the following figure supplement(s) for figure 6:

**Figure supplement 1.** Seroconversion associates with recovery of blood cell counts and hypoalbuminemia.

Other significant correlations revealed that seroconversion associates with markers of liver dysfunction and/or vascular leakage. Seroconverted patients showed elevated levels of blood urea nitrogen (BUN), elevated alkaline phosphatase (ALP), and decreased levels of albumin (ALB) (*Figure 6a,d,e*). Elevated BUN is indicative of liver and/or kidney malfunction. Elevated ALP is indicative of liver damage. Although the markers of liver injury AST (aspartate transaminase) and ALT (alanine transaminase) are elevated in the COVID19 cohort relative to the control group, they are not significantly associated with seroconversion status (*Figure 6a*, *Figure 6—figure supplement 1a,b*). Creatinine was significantly lower in COVID19 patients, also a sign of liver pathology, but no different by seroconversion status (*Figure 6—figure supplement 1a,b*). Low levels of circulating ALB, often reaching hypoalbuminemia, is a common feature of COVID19 pathology that has been associated with worse prognosis independently of age and comorbidities (*Huang et al., 2020*). It has been proposed that strong hypoalbuminemia without differences in AST and ALT could be due to inflammation-driven escape of serum ALB into interstitial space downstream of increased vascular permeability (*Huang et al., 2020*). In order to investigate this further, we probed the MS proteomics data set to see if other abundant liver-derived proteins showed similar behavior (*Supplementary file 2*). Indeed, seroconversion correlates with significantly decreased levels of major liver-derived proteins such as FETUB (fetuin B), PON1 (Paraoxonase 1), HPX (hemopexin), A1BG (alpha-1b-glycoprotein), APOA1 (apolipoprotein A1), and BCHE (butyrylcholinesterase) (*Figure 6—figure supplement 1c*, *Supplementary file 2*). This phenomenon could explain why many liver-derived factors involved in hemostasis control are also depleted in sero-high patients (e.g. fibrinogens, F2, F7, kallikreins) (*Figure 5a*, *Figure 5—figure supplement 1a,b*). Altogether, these results reveal that seroconversion is associated with recovery in diverse blood cell types, indicative of emergency hematopoiesis, along with biomarkers indicative of more severe liver dysfunction and/or increased vascular damage and interstitial leakage.

Lastly, we explored the possibility of defining a classifier that could discriminate the sero-low versus sero-high groups by using ratios of clinical laboratory values elevated upon seroconversion with ALB (which is decreased upon seroconversion) as the denominator. Expectedly, the WBC/ALB, D-dimer/ALB, Platelets/ALB, ALP/ALB, ANC/ALB, BUN/ALB, and ALC/ALB ratios were all significantly higher in the sero-high group (*Figure 6f*, *Figure 6—figure supplement 1d,e*). Using cut-off values that would capture >70% of the sero-high group when used individually, we noted that using any two of these cut offs concurrently would provide >90% specificity with >75% sensitivity in identifying a sero-high patient (*Figure 6g*). Increasing the classifier criteria to require passing any three or any four of these ratio cut-offs further increases specificity, at the cost of sensitivity (*Figure 6g*). Although the clinical utility of this classifier would require validation efforts in much larger cohorts, it nonetheless illustrates the variable clinical presentation of COVID19 pathology along a quantitative spectrum of seroconversion.

## Discussion

The temporal sequence of seroconversion relative to the onset of COVID19 symptoms has been already established (*Chen et al., 2020*; *Seow et al., 2020*). Within 2 weeks of symptom onset, virus-specific antibodies start accumulating in the bloodstream (*Seow et al., 2020*). Circulating IgMs, IgAs, and IgGs against SARS-CoV-2 increase rapidly thereafter, followed by decay of IgMs and IgAs over time, while IgGs remain high for several weeks and months (*Seow et al., 2020*). Building upon this knowledge, our integrated analysis of biosignatures of seroconversion using IgG measurements against SARS-CoV-2 polypeptides in hospitalized COVID19 patients supports a model for staging COVID19 pathology into a distinct sequence of early and late events, referred hereto as Stage 1 and Stage 2 (*Figure 7*). In this model, a seroconversion index based on IgG levels would increase over the first 2–4 weeks after symptom onset, with concomitant changes in COVID19 pathophysiology.

In Stage 1, associated with the initial antiviral response, hospitalized patients carry low levels of SARS-CoV-2-specific IgGs, high levels of circulating IFN ligands and complement subunits, signatures of activated T cells, pDCs and monocytes, high levels of cytokine-producing NK cells, as well as overall depletion of lymphocytes, neutrophils, and platelets. In Stage 2, associated with development of humoral immunity, patients carry high levels of SARS-CoV-2-specific antibodies, near-baseline levels of IFN ligands, complement subunits, and activated T cells, no significant cytopenias, but clear signs of B cell differentiation, plasmablast accumulation, platelet degranulation markers,

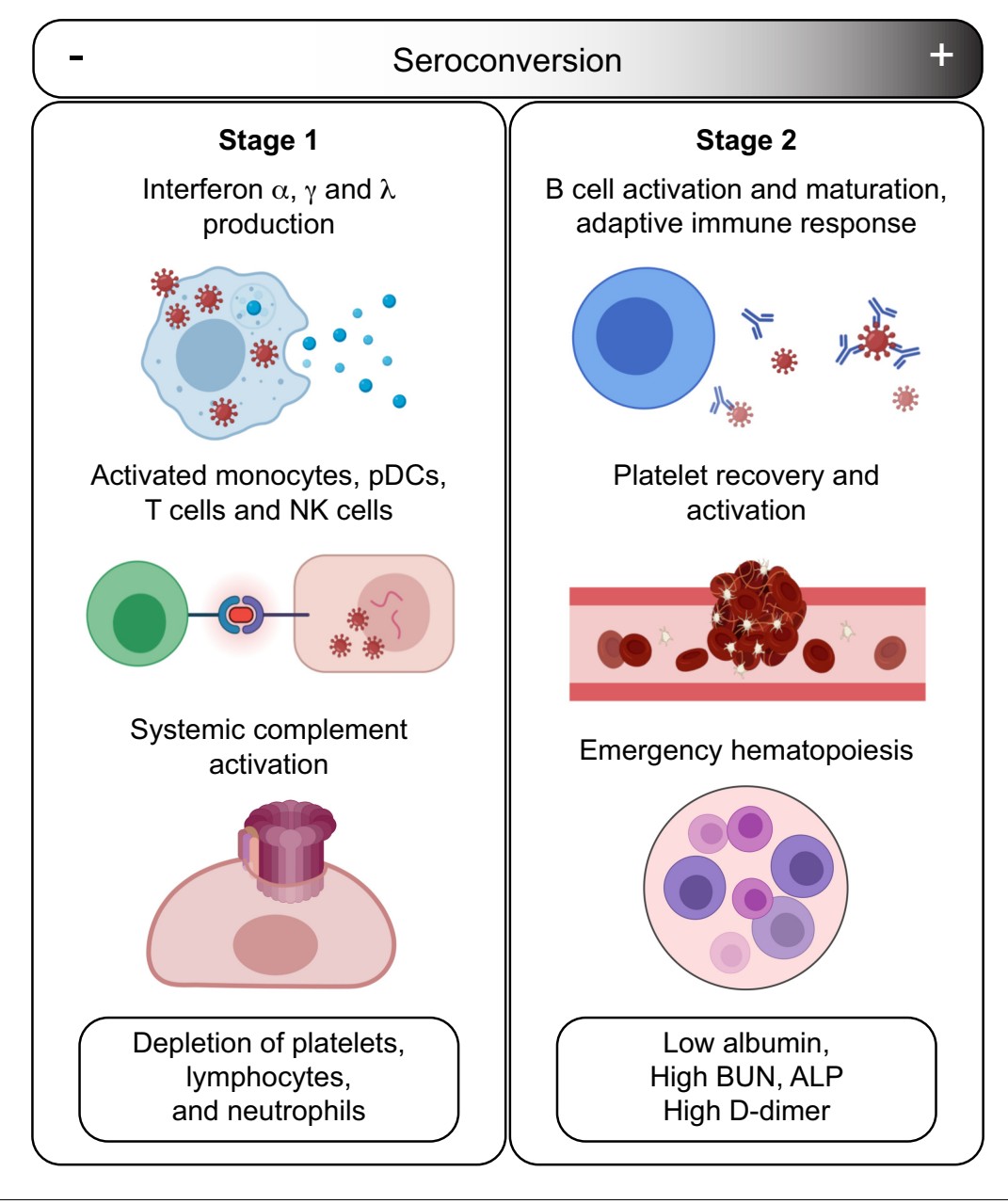

**Figure 7.** Model for staging COVID19 pathophysiology based on seroconversion status. Stage 1 applies to COVID19 patients with low degree of seroconversion and involves high levels of circulating IFNs, signs of strong systemic complement activation, hyperactive T cells, activated monocytes, and cytokine-producing NK cells, as well as depletion of key blood cell types. Stage 2 applies to COVID19 patients with high degree of seroconversion and is characterized by increased blood cell numbers, increased levels of markers of platelet degranulation, elevated D-dimer, and markers of increased liver dysfunction and/or interstitial leakage.

increased D-dimer levels, increased markers of liver damage (ALP, BUN), and depletion of circulating albumin.

Based on our findings, we propose the following sequence of events in mild-to-moderate symptomatic, hospitalized COVID19 patients. In Stage 1, COVID19 pathology is dominated by IFN- and complement-driven processes. Although SARS coronaviruses have evolved diverse strategies to evade the antiviral effects of IFNs (*Kindler et al., 2016*), IFN signaling nonetheless remains a potent defense mechanism against SARS-CoV-2, as illustrated by the fact that both genetic variants

compromising IFN signaling and autoantibodies against IFN ligands have been associated with severe COVID19 (*Bastard et al., 2020*; *Zhang et al., 2020*). However, sustained high levels of Type I and Type III IFN ligands could potentially contribute to SARS pathology through disruption of lung barrier function and other mechanisms (*Broggi et al., 2020*; *Major et al., 2020*). In mouse models of both SARS-CoV-1 and SARS-CoV-2 infections, Type I IFN signaling was shown to be required for development of lung pathology (*Israelow et al., 2020*; *Channappanavar et al., 2016*). In Stage 1, high IFN signaling is likely to drive activation of monocytes and T cells as well as B cell differentiation. In our studies, levels of IFNA2 are clearly correlated with activation of both CD4+ and CD8+ T cell subsets, with polarization of CD4+ T cells toward Th1 and Th17 states. Stage 1 is also correlated with signs of monocyte activation, demarked by high levels of MCP2/CCL8, CSF3, and circulating CD14, as well as high production of TNFSF13B. Stage 1 is also characterized by high levels of circulating complement factors. Recent studies have shown direct activation of the alternative complement pathway by the SARS-CoV-2 spike protein (*Yu et al., 2020*), and systemic complement activation has been associated with severe respiratory failure in COVID19 (*Holter et al., 2020*; *Carvelli et al., 2020*). Sustained high levels of complement activation could contribute to pathology through both the cytolytic effects of the MAC and the proinflammatory effects of the C5a-C5aR1 axis (*Carvelli et al., 2020*). Indeed, in a mouse model of SARS-CoV-1, depletion of the C3 complement subunit attenuated SARS pathology (*Gralinski et al., 2018*), and anti-C5aR1 neutralizing antibodies inhibited the C5a-mediated recruitment of human myeloid cells to lung tissue and reduced accompanying lung injury in a humanized mouse model (*Carvelli et al., 2020*). Complement could damage the endothelial tissue in the lung and other organs, compromising vascular integrity. Of note, early systemic elevation of complement subunits followed by elevation of the complement inhibitor SERPING1 has been described in the transition from pre-clinical to active tuberculosis (*Lubbers et al., 2020*; *Esmail et al., 2018*). Furthermore, markers elevated in Stage 1, such as CRP and CXCL10/IP-10, have been found to have high sensitivity and moderate specificity to triage patients with symptoms suggestive of active tuberculosis (*Santos et al., 2019*).

In this model, the transition from Stage 1 to Stage 2 is mediated by development of humoral immunity, as well as emergency hematopoiesis. As B cells differentiate, mature, and are selected by clonal evolution, plasmablasts are enriched in the bloodstream, producing specific IgGs of ever greater neutralizing capacity recognizing the RBD region of the spike protein. Seroconversion not only prevents viral reentry into cells, but could also prevent direct activation of the complement cascade by the spike protein (*Yu et al., 2020*). As humoral immunity develops, levels of IFN signaling and complement activity plummet. Increased levels of broad stimulators of hematopoiesis in Stage 1 (e.g. CSF3) would lead to emergency hematopoiesis. In turn, increased numbers of circulating platelets during Stage 2 would encounter the endothelial damage caused during Stage 1 by the cytolytic effects of the virus, disruption of lung barrier function by IFNs, complement attack, and even perhaps T cell- and monocyte-mediated cellular toxicity. In turn, the vascular damage incurred during Stage 1 would contribute to leakage of serum proteins in Stage 2 (i.e. hypoalbuminemia) and/or decreased production of abundant liver proteins due to increased liver damage. Although our analyses revealed significant remodeling of the hemostasis network in Stage 1 and Stage 2, we cannot conclusively interpret these results in terms of relative risk of thromboembolism and microangiopathy. Although D-dimer levels are higher in sero-high patients, our study is not powered to conclude if thrombotic disease is higher upon seroconversion.

According to this model, therapeutic interventions being tested in COVID19 could benefit from patient stratification based on a quantitative assessment of seroconversion. For example, JAK inhibitors and complement inhibitors, both of which have been tested for treatment of COVID19 pathology (*Kalil et al., 2020*; *Cantini et al., 2020*; *Mastellos et al., 2021*), may be more effective in cohorts enriched for sero-low patients. Clinical biomarkers for Stage 1 could be high IFN levels, high complement levels, and/or severe cytopenia. In contrast, patients in Stage 2 may not benefit from JAK inhibitors and complement inhibitors. Instead, they could benefit from better management of liver dysfunction and/or vascular damage. Notably, albumin supplementation was found to improve oxygenation in ARDS (*Uhlig et al., 2014*). This model also indicates that analysis of clinical trial data for various therapeutic interventions should take into consideration seroconversion status, as results could vary between Stage 1 and Stage 2 patients.

Given the cross-sectional nature of our study, our model should be challenged with longitudinal analysis of the biosignatures reported here. Our study does not address differences between

patients with mild, moderate, and severe disease, and our data does not include analysis of patients admitted to ICU with life-threatening COVID19. Nevertheless, our model indicates that time span between exposure to virus and seroconversion could be a key determinant of disease severity. The longer an affected individual remains in Stage 1, the more likely that the potential harmful effects of IFN hyperactivity and complement toxicity will be manifested, with increased likelihood of endothelial and organ damage, thus setting the stage for more severe thrombotic events and vascular damage in Stage 2. Notably, markers of disease severity could be found in either stage. For example, high IFNA2 (Stage 1 marker) and strong hypoalbuminemia (Stage 2 marker) have been independently associated with risk of severe COVID19 (*Huang et al., 2020*; *Lucas et al., 2020*). A delay in seroconversion could potentially explain in part the high risk of severe COVID19 pathology in the elderly, as B cell function and development of humoral immunity are decreased with age (*Frasca et al., 2011*). Importantly, an analysis of ~2M COVID19 cases in the USA disclosed during review of this manuscript confirms and expands on key aspects of our staging model, such as decreased levels of key inflammatory markers later in the course of hospitalization, along with significantly increased levels of WBCs and D-dimer (*Bennett et al., 2021*). Furthermore, this analysis supports the notion that disease severity is associated with increased and more prolonged inflammation in Stage 1 (as assessed by CRP levels) along with higher levels of D-dimer later on (*Bennett et al., 2021*).

In sum, our results support the existence of distinct pathophysiological states among hospitalized COVID19 patients, with seroconversion status being potentially useful as a surrogate marker of underlying processes. We hope these results will prompt additional investigations into the sequence of pathological events in COVID19 and how to ameliorate them for therapeutic purposes.

# Materials and methods

## Key resources table

| Reagent type (species) or resource | Designation | Source or reference | Identifiers | Additional information |
|---|---|---|---|---|
| Antibody | Anti-Human CD45 | Fluidigm | Cat# 3089003B, RRID:AB_2661851 | Monoclonal-Clone: HI30 Dilution: 1/200 |
| Antibody | Anti-Human CD57 | Biolegend | Cat# 322302, RRID:AB_2661815 | Mouse-Monoclonal-Clone: HCD57 Dilution: 1/100 |
| Antibody | Anti-Human CD11c | BD bioscience | Cat# 555390, RRID:AB_395791 | Mouse-Monoclonal-Clone: B-ly6 Dilution: 1/100 |
| Antibody | Anti-Human CD16 | eBioscience | Cat# 16-0167-85, RRID:AB_11040983 | Mouse-Monoclonal-Clone: B73.1 Dilution: 1/50 |
| Antibody | Anti-Human CD196 (CCR6) | Biolegend | Cat# 353402, RRID:AB_10918625 | Mouse-Monoclonal-Clone: 11a9 Dilution: 1/50 Stain live |
| Antibody | Anti-Human CD19 | Fluidigm | Cat# 3142001B, RRID:AB_2651155 | Monoclonal-Clone: HIB19 Dilution: 1/100 |
| Antibody | Anti-Human CD123 | Fluidigm | Cat# 3143014B, RRID:AB_2811081 | Mouse-Monoclonal-Clone: 6H6 Dilution: 1/123 |
| Antibody | Anti-Human CCR5 | Fluidigm | Cat# 3144007A | Monoclonal-Clone: NP6G4 Dilution: 1/25 Stain live |
| Antibody | Anti-Human IgD | Fluidigm | Cat# 3146005B, RRID:AB_2811082 | Mouse-Monoclonal-Clone: IA6-2 Dilution: 1/100 |
| Antibody | Anti-Human CD1c | Miltenyi | Cat# 130-108-032, RRID: AB_2661165 | Mouse-Monoclonal-Clone: AD5-8E7 Dilution: 1/30 |
| Antibody | Anti-Human CD38 | Biolegend | Cat# 303502, RRID:AB_314354 | Mouse-Monoclonal-Clone: HIT2 Dilution: 1/50 |
| Antibody | Anti-Human CD127 | Fluidigm | Cat# 3149011B, RRID:AB_2661792 | Monoclonal-Clone: A019D5 Dilution: 1/100 Stain live |

*Continued on next page*

*Continued*

| Reagent type (species) or resource | Designation | Source or reference | Identifiers | Additional information |
|---|---|---|---|---|
| Antibody | Anti-Human CD86 | Fluidigm | Cat# 3150020B, RRID:AB_2687852 | Monoclonal-Clone: IT2.2 Dilution: 1/100 |
| Antibody | Anti-Human ICOS | Biolegend | Cat# 313502, RRID:AB_416326 | Armenian hamster -Monoclonal-Clone: DX29 Dilution: 1/50 |
| Antibody | Anti-Human CD141 | Biolegend | Cat# 344102, RRID:AB_2201808 | Mouse-Monoclonal-Clone: M80 Dilution: 1/50 |
| Antibody | Anti-Human Tim3 | Fluidigm | Cat# 3153008B, RRID:AB_2687644 | Monoclonal-Clone: MBSA43 Dilution: 1/100 |
| Antibody | Anti-Human TIGIT | Fluidigm | Cat# 3154016B, RRID:AB_2888926 | Mouse-Monoclonal-Clone: F38-2E2 Dilution: 1/50 Stain live |
| Antibody | Anti-Human CD27 | Fluidigm | Cat# 3155001B, RRID:AB_2687645 | Mouse-Monoclonal-Clone: L128 Dilution: 1/100 |
| Antibody | Anti-Human CXCR3 | Fluidigm | Cat# 3156004B, RRID:AB_2687646 | Monoclonal-Clone: G025H7 Dilution: 1/100 Stain live |
| Antibody | Anti-Human CD45RA | Biolegend | Cat# 304102, RRID:AB_314406 | Mouse-Monoclonal-Clone: HI100 Dilution: 1/50 |
| Antibody | Anti-Human PD-1 | Biolegend | Cat# 329941, RRID:AB_2563734 | Mouse-Monoclonal-Clone: EH12.2H7 Dilution: 1/50 |
| Antibody | Anti-Human PDL1 | Fluidigm | Cat# 3159029B, RRID:AB_2861413 | Mouse-Monoclonal-Clone: 29E.2A3 Dilution: 1/100 |
| Antibody | Anti-Human CD14 | Fluidigm | Cat# 3160001B, RRID:AB_2687634 | Monoclonal-Clone: M5E2 Dilution: 1/100 |
| Antibody | Anti-Human Tbet | Fluidigm | Cat# 3161014B, RRID:AB_2858233 | Monoclonal-Clone: 4b10 Dilution: 1/100 |
| Antibody | Anti-Human Ki67 | Fluidigm | Cat# 3162012B, RRID:AB_2888928 | Mouse-Monoclonal-Clone: B56 Dilution: 1/100 |
| Antibody | Anti-Human CD33 | Fluidigm | Cat# 3163023B, RRID:AB_2687857 | Monoclonal-Clone: WM53 Dilution: 1/100 |
| Antibody | Anti-Human CD95 | Fluidigm | Cat# 3164008B, RRID:AB_2858235 | Monoclonal-Clone: DX2 Dilution: |
| Antibody | Anti-Human Foxp3 | Biolegend | Cat# 14-4774-82, RRID:AB_467552 | Mouse-Monoclonal-Clone: 150D/E4 Dilution: 1/50 |
| Antibody | Anti-Human Eomes | Biolegend | Cat# 14-4877-82, RRID:AB_2572882 | Mouse-Monoclonal-Clone: WD1928 Dilution: 1/100 |
| Antibody | Anti-Human CCR7 | Fluidigm | Cat# 3167009A, RRID:AB_2858236 | Monoclonal-Clone: G043H7 Dilution: 1/100Stain live |
| Antibody | Anti-Human CD8a | Fluidigm | Cat# 3168002B | Monoclonal-Clone: SK1 Dilution: 1/100 |
| Antibody | Anti-Human CD25 | Fluidigm | Cat# 3169003B, RRID:AB_2661806 | Monoclonal-Clone: 2A3 Dilution: 1/100 Stain live |
| Antibody | Anti-Human CD3 | Fluidigm | Cat# 3170001B, RRID:AB_2811085 | Mouse-Monoclonal-Clone: UCHT1 Dilution: 1/100 |
| Antibody | Anti-Human CXCR5 | Fluidigm | Cat# 3171014B, RRID:AB_2858239 | Monoclonal-Clone: 51505 Dilution: 1/100 Stain live |
| Antibody | Anti-Human IgM | Fluidigm | Cat# 3172004B, RRID:AB_2810858 | Mouse-Monoclonal-Clone: MHM-88 Dilution: 1/100 Stain live |
| Antibody | Anti-Human HLA-DR | Fluidigm | Cat# 3173005B, RRID:AB_2810248 | Monoclonal-Clone: L243 Dilution: 1/100 |
| Antibody | Anti-Human CD4 | Fluidigm | Cat# 3174004B, RRID:AB_2687862 | Monoclonal-Clone: SK3 Dilution: 1/100 |

*Continued*

| Reagent type (species) or resource | Designation | Source or reference | Identifiers | Additional information |
|---|---|---|---|---|
| Antibody | Anti-Human CCR4 | R and D | Cat# MAB1567-500 | Mouse-Monoclonal-Clone: 205410 Dilution: 1/50 Stain live |
| Antibody | Anti-Human CD56 | Miltenyi | Cat# 130-113-312, RRID:AB_2726090 | Monoclonal-Clone: HCD56 Dilution: 1/200 |
| Antibody | Anti-Human CD11b | Fluidigm | Cat# 3209003B, RRID:AB_2687654 | Monoclonal-Clone: ICRF44 Dilution: 1/200 |
| Commercial assay or kit | U-PLEX Biomarker Group 1 (hu) 71-Plex | Meso Scale Discovery (MSD) | Cat# K15081K | |
| Commercial assay or kit | V-PLEX Vascular Injury Panel 2 Human Kit | Meso Scale Discovery (MSD) | Cat# K15198D | |
| Commercial assay or kit | V-PLEX Angiogenesis Panel 1 Human Kit | Meso Scale Discovery (MSD) | Cat# K15190D | |
| Commercial assay or kit | PAXgene Blood RNA Tubes | PreAnalytiX/Qiagen | Cat# 762165 | |
| Commercial assay or kit | PAXgene Blood RNA Kit | Qiagen | Cat# 762164 | |
| Commercial assay or kit | Universal Plus mRNA-Seq with NuQuant, Human Globin AnyDeplete | Tecan | Cat# 0521-A01 | |
| Software, algorithm | R | R Foundation for Statistical Computing | v4.0.1 RRID:SCR_001905 | https://www.R-project.org/ |
| Software, algorithm | RStudio | RStudio, Inc | v1.3.959 RRID:SCR_000432 | http://www.rstudio.com/ |
| Software, algorithm | Bioconductor | N/A | v3.11 RRID:SCR_006442 | https://bioconductor.org/ |
| Software, algorithm | Tidyverse collection of packages for R | N/A | N/A RRID:SCR_019186 | https://www.tidyverse.org/ |
| Software, algorithm | limma package for R | N/A | v3.44.3 RRID:SCR_010943 | https://bioconductor.org/packages/release/bioc/html/limma.html |
| Software, algorithm | CellEngine | Primity Bio Inc | N/A | https://primitybio.com/cellengine.html |
| Software, algorithm | bcl2fastq | Illumina, Inc | v2.20.0.422 RRID:SCR_015058 | https://support.illumina.com/sequencing/sequencing_software/bcl2fastq-conversion-software.html |
| Software, algorithm | FASTQC | N/A | v0.11.5 RRID:SCR_014583 | https://www.bioinformatics.babraham.ac.uk/projects/fastqc/ |
| Software, algorithm | FastQ Screen | N/A | v0.11.0 RRID:SCR_000141 | https://www.bioinformatics.babraham.ac.uk/projects/fastq_screen/ |
| Software, algorithm | bbduk/BBTools | N/A | v37.99 RRID:SCR_016968 | https://jgi.doe.gov/data-and-tools/bbtools/ |
| Software, algorithm | fastq-mcf/ea-utils | N/A | v1.05 RRID:SCR_005553 | https://expressionanalysis.github.io/ea-utils/ |
| Software, algorithm | HISAT2 | N/A | v2.1.0 RRID:SCR_015530 | http://daehwankimlab.github.io/hisat2/ |
| Other | Human genome reference fasta | N/A | GRCh38 RRID:SCR_014966 | ftp://ftp.ebi.ac.uk/pub/databases/gencode/Gencode_human/release_33/GRCh38.primary_assembly.genome.fa.gz |
| Other | Human genome annotation GTF file | Gencode | v33 RRID:SCR_014966 | ftp://ftp.ebi.ac.uk/pub/databases/gencode/Gencode_human/release_33/gencode.v33.basic.annotation.gtf.gz |
| Software, algorithm | Samtools | N/A | v1.5 | http://www.htslib.org/ |
| Software, algorithm | HTSeq-count | N/A | v0.6.1 RRID:SCR_005514 | https://htseq.readthedocs.io/en/master/ |

*Continued on next page*

*Continued*

| Reagent type (species) or resource | Designation | Source or reference | Identifiers | Additional information |
|---|---|---|---|---|
| Software, algorithm | DESeq2 package for R | N/A | v1.28.1 RRID:SCR_015687 | https://bioconductor.org/packages/release/bioc/html/DESeq2.html |
| Software, algorithm | Hmisc package for R | N/A | v4.4–0 | https://cran.r-project.org/web/packages/Hmisc/index.html |
| Software, algorithm | ggplot2 package for R | N/A | v3.3.1 RRID:SCR_014601 | https://ggplot2.tidyverse.org/ |
| Software, algorithm | rstatix package for R | N/A | v0.6.0 | https://cran.r-project.org/web/packages/rstatix/index.html |
| Software, algorithm | ComplexHeatmap package for R | N/A | v2.4.2 RRID:SCR_017270 | https://www.bioconductor.org/packages/release/bioc/html/ComplexHeatmap.html |
| Software, algorithm | ggforce package for R | N/A | v0.3.1 | https://ggforce.data-imaginist.com/reference/index.html |

## Study design, participant recruitment, and clinical data capture

Research participants were recruited and consented for participation in the COVID Biobank of the University of Colorado Anschutz Medical Campus [Colorado Multiple Institutional Review Board (COMIRB) Protocol # 20–0685]. Data was generated from deidentified biospecimens and linked to demographics and clinical metadata procured through the Health Data Compass of the University of Colorado under COMIRB Protocol # 20–1700. Participants were hospitalized either at Children's Hospital Colorado or at the University of Colorado Hospital. COVID status was defined by a positive PCR result and/or antibody test within 14 days of the research blood draw. The control cohort consisted of COVID19-negative research participants receiving medical care for a range of conditions, none of them in critical condition at the time of the research blood draw. Cohort characteristics can be found in *Supplementary file 1*.

## Blood processing

Blood samples were collected into EDTA tubes, sodium heparin tubes, and PAXgene Blood RNA Tubes (PreAnalytiX/Qiagen). After centrifugation, EDTA plasma was used for MS proteomics, SOMAscan proteomics, as well as multiplex immunoassays using MSD technology for both cytokine profiles and seroconversion assays. From sodium heparin tubes, PBMCs were obtained by the Ficoll gradient method before cryopreservation and assembly of batches for MC analysis (see below).

## Plasma proteomics by mass spectrometry

Plasma samples were digested in S-Trap filters (Protifi, Huntington, NY) according to the manufacturer's procedure. Briefly, a dried protein pellet prepared from organic extraction of patient plasma was solubilized in 400 µL of 5% (w/v) SDS. Samples were reduced with 10 mM DTT at 55°C for 30 min, cooled to room temperature, and then alkylated with 25 mM iodoacetamide in the dark for 30 min. Next, a final concentration of 1.2% phosphoric acid and then six volumes of binding buffer [90% methanol; 100 mM triethylammonium bicarbonate (TEAB); pH 7.1] were added to each sample. After gentle mixing, the protein solution was loaded into an S-Trap filter, spun at 2000 rpm for 1 min, and the flow-through collected and reloaded onto the filter. This step was repeated three times, and then the filter was washed with 200 µL of binding buffer three times. Finally, 1 µg of sequencing-grade trypsin (Promega) and 150 µL of digestion buffer (50 mM TEAB) were added onto the filter and digestion carried out at 47°C for 1 hr. To elute peptides, three stepwise buffers were applied, 200 µL of each with one more repeat, including 50 mM TEAB, 0.2% formic acid (FA) in $H_2O$, and 50% acetonitrile and 0.2% FA in $H_2O$. The peptide solutions were pooled, lyophilized, and resuspended in 1 mL of 0.1% FA. 20 µL of each sample was loaded onto individual Evotips for desalting and then washed with 20 µL 0.1% FA followed by the addition of 100 µL storage solvent (0.1% FA) to keep the Evotips wet until analysis. The Evosep One system (Evosep, Odense, Denmark) was used to separate peptides on a Pepsep column, (150 µm internal diameter, 15 cm) packed with ReproSil C18 1.9 µm, 120A resin. The system was coupled with a timsTOF Pro mass

spectrometer (Bruker Daltonics, Bremen, Germany) via a nano-electrospray ion source (Captive Spray, Bruker Daltonics). The mass spectrometer was operated in PASEF mode. The ramp time was set to 100 ms and 10 PASEF MS/MS scans per topN acquisition cycle were acquired. MS and MS/MS spectra were recorded from *m/z* 100 to 1700. The ion mobility was scanned from 0.7 to 1.50 Vs/cm$^2$. Precursors for data-dependent acquisition were isolated within ±1 Th and fragmented with an ion mobility-dependent collision energy, which was linearly increased from 20 to 59 eV in positive mode. Low-abundance precursor ions with an intensity above a threshold of 500 counts but below a target value of 20000 counts were repeatedly scheduled and otherwise dynamically excluded for 0.4 min. Raw data file conversion to peak lists in the MGF format, downstream identification, validation, filtering, and quantification were managed using FragPipe version 13.0. MSFragger version 3.0 was used for database searches against a Human isoform-containing UniProt fasta file (version 08/11/2020) with decoys and common contaminants added. The identification settings were as follows: Trypsin, Specific, with a maximum of two missed cleavages, up to two isotope errors in precursor selection allowed for, 10.0 ppm as MS1 and 20.0 ppm as MS2 tolerances; fixed modifications: Carbamidomethylation of C (+57.021464 Da), variable modifications: Oxidation of M (+15.994915 Da), Acetylation of protein N-term (+42.010565 Da), Pyrolidone from peptide N-term Q or C (−17.026549 Da). The Philosopher toolkit version 3.2.9 (build 1593192429) was used for filtering of results at the peptide and protein level at 0.01 FDR. Label-free quantification was performed by AUC integration with matching between all runs using IonQuant.

## Plasma proteomics by SOMAscan assays

125 μL EDTA plasma was analyzed by SOMAscan assays using previously established protocols (*Gold et al., 2012*). Briefly, each of the 5000+ SOMAmer reagents binds a target peptide and is quantified on a custom Agilent hybridization chip. Normalization and calibration were performed according to SOMAscan Data Standardization and File Specification Technical Note (SSM-020) (*Gold et al., 2012*). The output of the SOMAscan assay is reported in relative fluorescent units (RFU).

## Cytokine profiling and seroconversion by multiplex immunoassay

Multiplex immunoassays assays were performed on EDTA plasma aliquots following manufacturer's instructions (Meso Scale Discovery, MSD). A list of immune factors measured by MSD can be found in *Supplementary file 4*. Values were extrapolated against a standard curve using provided calibrators. Seroconversion assays against SARS-CoV-2 proteins and the control protein from the Flu A Hong Kong H3 virus were performed in a multiplex immunoassay using the IgG detection readout according to manufacturer's instructions (MSD). Relative values were extrapolated against a standardized curve consisting of pooled COVID19 positive reference plasma (*Johnson et al., 2020*).

## Mass cytometry analysis of immune cell types

Cryopreserved PBMCs were thawed, washed twice with Cell Staining Buffer (CSB) (Fluidigm), and counted with an automated cell counter (Countess II , Thermo Fisher Scientific). Extracellular staining of live cells was done in CSB for 30 min at room temperature, in 3–5 x 10$^6$ cells per sample. Cells were washed with 1× PBS (Fluidigm) and stained with 1 mL of 0.25 mM cisplatin (Fluidigm) for 1 min at room temperature for exclusion of dead cells. Samples were then washed with CSB and incubated with 1.6% PFA (Electron Microscopy Sciences) for 10 min at room temperature. Samples were washed with CBS and barcoded using a Cell-IDTM 20-Plex Pd Barcoding Kit (Fluidigm) of lanthanide-tagged cell reactive metal chelators to covalently label samples with a unique combination of palladium isotopes, then combined. Surface staining with antibodies that work on fixed epitopes was performed in CSB for 30 min at room temperature (see *Supplementary file 11* and Key resources table for antibody information). Cells were washed twice with CSB and fixed in Fix/Perm buffer (eBioscience) for 30 min, washed in permeabilization buffer (eBioscience) twice, then intracellular factors were stained in permeabilization buffer for 45 min at 4˚C. Cells were washed twice with Fix/Perm Buffer and were labeled overnight at 4˚C with Cell-ID Intercalator-Ir (Fluidigm) for DNA staining. Cells were then analyzed on a Helios instrument (Fluidigm). To make all samples comparable, pre-processing of MCmass cytometry data included normalization within and between batches via polystyrene beads embedded with lanthanides as previously described (*Finck et al., 2013*). Files

were debarcoded using the Matlab DebarcoderTool (*Zunder et al., 2015*). Then normalization again between batches relative to a reference batch based on technical replicates (*Schuyler et al., 2019*). Gating was performed using CellEngine (Primitybio). Gating strategy is summarized in *Figure 1—figure supplement 1c–g* and *Supplementary file 12*.

## Whole-blood RNA library preparation and sequencing

RNA was purified from PAXgene Blood RNA Tubes (PreAnalytiX/Qiagen) using a PAXgene Blood RNA Kit (Qiagen), according to the manufacturer's instructions. RNA quality was assessed using an Agilent 2200 TapeStation and quantified by Qubit (Life Technologies). Globin RNA depletion, poly-A(+) RNA enrichment, and strand-specific library preparation were carried out using a Universal Plus mRNA-Seq with NuQuant, Human Globin AnyDeplete (Tecan). Paired-end 150 bp sequencing was carried out on an Illumina NovaSeq 6000 instrument by the Genomics Shared Resource at the University of Colorado Anschutz Medical Campus.

## Biostatistics and bioinformatics analyses

Preprocessing, statistical analysis, and plot generation for all data sets were carried out using R (R 4.0.1/RStudio 1.3.959/Bioconductor v 3.11) (*Huber et al., 2015*; *R Development Core Team, 2020*; *Team R Studio, 2020*), as detailed below.

### MSD seroconversion data

Plasma concentration values (pg/mL) for IgGs recognizing SARS-Co-V-2 and Flu A Hong Kong H3 epitopes were adjusted for sex and age using the *removeBatchEffect* function from the limma package (v 3.44.3) (*Ritchie et al., 2015*). Distributions of sex/age-adjusted concentration values for each epitope in COVID19 positive and COVID19 negative samples were compared using the Wilcoxon–Mann–Whitney two-sample rank-sum test, with Benjamini–Hochberg correction of p-values and an estimated false discovery rate (FDR) threshold of 0.1 ($q < 0.1$). To capture seroconversion as a single value we calculated a 'seroconversion index' for each sample as follows. First, Z-scores were calculated from the adjusted concentration values for each epitope in each sample, based on the mean and standard deviation of COVID19-negative samples. Second, the per-sample seroconversion index was calculated as the sum of Z-scores for the four SARS-CoV-2 seroconversion assays. For comparison of multiple measurements from COVID19 positive samples with high seroconversion indices to those with low seroconversion indices, or COVID19 negative samples, COVID19 positive samples were divided into two equal-sized groups based on their seroconversion index, referred to as 'sero-low' versus 'sero-high' groups.

### Mass spectrometry proteomics data

Raw Razor intensity data were filtered for high abundance proteins by removing those with >70% zero values in both COVID19 negative and COVID19 positive groups. For the remaining 407 abundant proteins, 0 values (8363 missing values of 44,363 total measurements) were replaced with a random value sampled between 0 and $0.5\times$ the minimum nonzero intensity value for that protein. Data was then normalized using a scaling factor derived from the global median intensity value across all proteins/sample median intensity across all proteins (*De Livera et al., 2012*) and adjusted for sex and age using the *removeBatchEffect* function from the limma package (v 3.44.3) (*Ritchie et al., 2015*).

### SOMAscan proteomics data

Normalized data (RFU) was imported and converted from a SOMAscan.adat file using a custom R package (SomaDataIO) and adjusted for sex and age using the *removeBatchEffect* function from the limma package (v 3.44.3) (*Ritchie et al., 2015*).

### MSD cytokine profiling data

Plasma concentration values (pg/mL) for each of the cytokines and related immune factors measured across multiple MSD assay plates was imported to R, combined, and analytes with >10% of values outside of detection or fit curve range flagged. For each analyte, missing values were replaced with either the minimum (if below fit curve range) or maximum (if above fit curve range) calculated

concentration and means of duplicate wells used in all further analysis. Data was adjusted for sex and age using the *removeBatchEffect* function from the limma package (v 3.44.3) (*Ritchie et al., 2015*).

### Mass cytometry data
Cell population frequencies, exported from CellEngine as percentages of various parental lineages, were adjusted for sex and age using the *removeBatchEffect* function from the limma package (v 3.44.3) (*Ritchie et al., 2015*).

### RNA-seq data
RNA-seq data yield was ~40–80 × $10^6$ raw reads and ~32–71 × $10^6$ final mapped reads per sample. Reads were demultiplexed and converted to fastq format using bcl2fastq (bcl2fastq v2.20.0.422). Data quality was assessed using FASTQC (v0.11.5) (https://www.bioinformatics.babraham.ac.uk/projects/fastqc/) and FastQ Screen (v0.11.0, https://www.bioinformatics.babraham.ac.uk/projects/fastq_screen/). Trimming and filtering of low-quality reads were performed using bbduk from BBTools (v37.99)(*Bushnell et al., 2017*) and fastq-mcf from ea-utils (v1.05, https://expressionanalysis.github.io/ea-utils/). Alignment to the human reference genome (GRCh38) was carried out using HISAT2 (v2.1.0)(*Kim et al., 2019*) in paired, spliced-alignment mode with a GRCh38 index with a Gencode v33 annotation GTF, and alignments were sorted and filtered for mapping quality (MAPQ > 10) using Samtools (v1.5)(*Li et al., 2009*). Gene-level count data were quantified using HTSeq-count (v0.6.1)(*Anders et al., 2015*) with the following options (–stranded=reverse –minaqual=10 –type=exon –mode=intersection-nonempty) using a Gencode v33 GTF annotation file. Differential gene expression (COVID-positive versus COVID-negative) was evaluated using DESeq2 (version 1.28.1)(*Love et al., 2014*) in R (version 4.0.1), using q < 0.1 (FDR < 10%) as the threshold for differentially expressed genes.

### Correlation analysis
To identify features in each data set that correlate with Seroconversion Index in COVID19 positive samples, Spearman *rho* values and p-values were calculated against the sex/age-adjusted values for each data set using the *rcorr* function from the Hmisc package (v 4.4–0) (*Feh, 2020*), with Benjamini–Hochberg correction of p-values and an estimated FDR threshold of 0.1. For visualization, XY scatter plots with points colored by local density were generated using a custom density function and the ggplot2 (v3.3.1) package (*Wickham, 2016*).

### Comparison of seroconversion groups
Distributions of sex/age-adjusted concentration values for features in COVID19-negative samples and COVID19-positive samples with low vs. high seroconversion indices were compared with pairwise Wilcoxon–Mann–Whitney two-sample rank-sum tests, using the *wilcox_test* function from the rstatix package (v0.6.0) (*Kassambara, 2020*), with Benjamini–Hochberg correction of p-values and an estimated FDR threshold of 0.1 (q < 0.1). To visualize the differences between COVID negative samples and COVID positive samples with low vs. high seroconversion indices, Z-scores were calculated for each feature based on the mean and standard deviation of COVID-negative samples, and visualized as heatmaps and/or modified sina plots using the ComplexHeatmap (v2.4.2) (*Gu et al., 2016*), ggplot2 (v3.3.1), and ggforce (v0.3.1) packages (*Pedersen, 2019*).

### Sample size estimates and replicates
Sample size was estimated based on previous published studies investigating autoinflammatory processes in individuals with Down syndrome using technologies identical to those employed in this study, such as cytokine profiling using multiplex immunoassays (*Sullivan et al., 2017*; *Araya et al., 2019*; *Powers et al., 2019*); SOMAscan proteomics (*Sullivan et al., 2017*); and RNA-seq transcriptome analysis (*Araya et al., 2019*; *Powers et al., 2019*; *Sullivan et al., 2016*; *Waugh et al., 2019*). A formal power analysis was not performed before deciding on sample size for this study. Instead, based on our studies of interferon-driven inflammation in Down syndrome, and assuming similar or greater size effects in COVID19, we estimated that a sample size of 30+ controls versus 70+ COVID19 patients would suffice to identify statistically significant changes in cytokines, plasma

proteins, mRNAs, and metabolites. Seroconversion assays were performed in duplicate for each plasma sample, multiplexed immunoassays for cytokines were performed in duplicate for each plasma sample, MC immune mapping was performed once for each individual fraction of PBMCs, SOMAscan proteomics was completed once for each plasma sample, RNAseq was performed once for each whole blood RNA sample, and mass spectrometry proteomics was performed once for each plasma sample. Each research participant is considered a biological replicate for the purpose of the comparisons in this study, such as COVID19 negative versus COVID19 positive, or sero-low versus sero-high groups among COVID19 patients. Extreme outlier data points (above $Q3 + 3xIQR$ or below $Q1 - 3xIQR$) were removed.

## Acknowledgements

We are grateful to Dr. Thomas Flaig and the Office of the Vice Chancellor For Research (OVCR) at the University of Colorado Anschutz Medical Campus for their leadership in setting up the COVID19 Biobank at the University of Colorado and also to the COVID19 Biobank Steering Committee for overall support of this project. We thank the members of the Biorepository Shared Resource, especially Dr. Adrie Van Bokhoven, Zachary Grasmick, and Hannah Schumman; members of the Human Immune Monitoring Shared Resource, especially Dr. Jill Slansky, Jodi Livesay, Troy Schedin, and Jennifer McWilliams; members of the Flow Cytometry Shared Resource of the University of Colorado Cancer Center, specially Eric Cambley, Alistair Acosta, Christine Childs, and Kristina Terrell; as well as Aaron Issaian for assistance with MS proteomics data analysis. We also thank the SomaLogic team for their support and the Meso Scale Discovery team for generous support with seroconversion assays. We are grateful to Dr. Ian Brooks, Michele Edelmann, and the rest of the Health Data Compass team for the clinical data.

## Additional information

### Competing interests

Joaquín M Espinosa: Reviewing editor, *eLife,* is a co-inventor on two patents related to JAK inhibition in COVID19: U.S. Provisional Patent Application Serial No. 62/992,855 entitled 'JAK1 Inhibition For Modulation Of Overdrive Anti-Viral Response To COVID-19'; U.S. Provisional Patent Application Serial No. 62/993,749 entitled 'Compounds and Methods for Inhibition or Modulation of Viral Hypercytokinemia'. Kelly D Sullivan: is a co-inventor on two patents related to JAK inhibition in COVID19: U.S. Provisional Patent Application Serial No. 62/992,855 entitled 'JAK1 Inhibition For Modulation Of Overdrive Anti-Viral Response To COVID-19'; U.S. Provisional Patent Application Serial No. 62/993,749 entitled 'Compounds and Methods for Inhibition or Modulation of Viral Hypercytokinemia'. The other authors declare that no competing interests exist.

### Funding

| Funder | Grant reference number | Author |
| --- | --- | --- |
| National Institute of Allergy and Infectious Diseases | R01AI150305 | Joaquín M Espinosa |
| National Institute of Allergy and Infectious Diseases | 3R01AI150305-01S1 | Joaquín M Espinosa |
| National Center for Advancing Translational Sciences | UL1TR002535 | Tellen D Benett Joaquín M Espinosa |
| National Center for Advancing Translational Sciences | 3UL1TR002535-03S2 | Tellen D Benett |
| National Heart, Lung, and Blood Institute | R01HL149714 | Angelo D'Alessandro |
| National Heart, Lung, and Blood Institute | R01HL148151 | Angelo D'Alessandro |
| National Heart, Lung, and Blood Institute | R21HL150032 | Angelo D'Alessandro |

| National Cancer Institute | P30CA046934 | Angelo D'Alessandro Kirk C Hansen Joaquín M Espinosa |
|---|---|---|
| National Institute of General Medical Sciences | R35GM124939 | Andrew A Monte |
| National Institute of General Medical Sciences | RM1GM131968 | Angelo D'Alessandro |
| National Institute of Allergy and Infectious Diseases | R01AI145988 | Kelly D Sullivan |
| Global Down Syndrome Foundation | | Kelly D Sullivan Joaquín M Espinosa |
| Anna and John J Sie Foundation | | Kelly D Sullivan Joaquín M Espinosa |
| Boettcher Foundation | | Kelly D Sullivan Elena WY Hsieh |
| Lyda Hill Foundation | | Tellen D Benett Joaquín M Espinosa |

The funders had no role in study design, data collection and interpretation, or the decision to submit the work for publication.

## Author contributions

Matthew D Galbraith, Conceptualization, Data curation, Formal analysis, Validation, Investigation, Visualization, Methodology, Writing - original draft, Writing - review and editing; Kohl T Kinning, Data curation, Formal analysis, Investigation, Visualization, Writing - review and editing; Kelly D Sullivan, Conceptualization, Formal analysis, Investigation, Writing - review and editing; Ryan Baxter, Data curation, Formal analysis, Investigation, Writing - review and editing; Paula Araya, Conceptualization, Data curation, Formal analysis, Investigation, Visualization, Methodology, Writing - review and editing; Kimberly R Jordan, Data curation, Formal analysis, Supervision, Funding acquisition, Investigation, Methodology, Writing - review and editing; Seth Russell, Jessica R Shaw, Monika Dzieciatkowska, Data curation, Formal analysis, Investigation, Methodology, Writing - review and editing; Keith P Smith, Data curation, Investigation, Methodology, Writing - review and editing; Ross E Granrath, Investigation, Methodology, Writing - review and editing; Tusharkanti Ghosh, Data curation, Formal analysis, Methodology, Writing - review and editing; Andrew A Monte, Formal analysis, Investigation, Methodology, Writing - review and editing; Angelo D'Alessandro, Conceptualization, Data curation, Formal analysis, Supervision, Funding acquisition, Investigation, Methodology, Writing - review and editing; Kirk C Hansen, Elena WY Hsieh, Conceptualization, Data curation, Formal analysis, Supervision, Funding acquisition, Investigation, Visualization, Methodology, Writing - review and editing; Tellen D Benett, Conceptualization, Formal analysis, Supervision, Funding acquisition, Investigation, Visualization, Methodology, Writing - review and editing; Joaquín M Espinosa, Conceptualization, Formal analysis, Supervision, Funding acquisition, Investigation, Visualization, Methodology, Writing - original draft, Project administration

## Author ORCIDs

Matthew D Galbraith (iD) https://orcid.org/0000-0003-0485-3927
Kelly D Sullivan (iD) https://orcid.org/0000-0003-2725-0205
Seth Russell (iD) http://orcid.org/0000-0002-2436-1367
Joaquín M Espinosa (iD) https://orcid.org/0000-0001-9048-1941

## Ethics

Human subjects: Research participants were recruited and consented for participation in the COVID Biobank of the University of Colorado Anschutz Medical Campus [Colorado Multiple Institutional Review Board (COMIRB) Protocol # 20-0685]. Data was generated from deidentified biospecimens and linked to demographics and clinical metadata procured through the Health Data Compass of the University of Colorado under COMIRB Protocol # 20-1700.

Decision letter and Author response
Decision letter https://doi.org/10.7554/eLife.65508.sa1
Author response https://doi.org/10.7554/eLife.65508.sa2

## Additional files

#### Supplementary files

• Supplementary file 1. Cohort characteristics. Table summarizing cohort characteristics. Information pertaining to less than 10 participants is indicated as <10 to prevent potential reidentification. For clinical labs, values represent the mean ± standard deviation. Acronyms for clinical laboratory measurements are as follows: BUN: blood urea nitrogen; CRP: C-reactive protein; ALT: alanine aminotransferase; ALP: alkaline phosphatase; AST: aspartate aminotransferase; BNP: brain natriuretic peptide. Comorbidities affecting each group are listed based on two different annotations: Carlson and Elixhauser. Acronyms for comorbidities are as follows: CHF: chronic heart failure; DM: diabetes mellitus; DMCX: diabetes with complications; METS: metastatic cancer; MI: myocardial infarction; PUD: peptic ulcer disease; PVD: peripheral vascular disease; HTN: hypertension; PHTN: pulmonary hypertension. Fisher's exact test was used to calculate p-values for differences in % among groups, and the Mann–Whitney test was used to calculate p-values for differences in clinical lab values.

• Supplementary file 2. Seroconversion versus mass spectrometry (MS) proteomics. Results of Spearman correlation analysis between seroconversion indices and proteins identified by MS. Column A indicates the protein name, column B indicates the SwissProt ID, column C indicates the Spearman *rho* value, column D indicates the p-value, and column E indicates the adjusted p-value using the Benjamini–Hochberg method.

• Supplementary file 3. Seroconversion versus SOMAcan proteomics. Results of Spearman correlation analysis between seroconversion indices and proteins identified by SOMAscan technology. Column A indicates the SOMAmer identification number, column B indicates the target protein recognized by the SOMAmer, column C indicates the SwissProt ID, column D indicates the gene symsol, column E indicates the Spearman *rho* value, column F indicates the p-value, and column G indicates the adjusted p-value using the Benjamini–Hochberg method.

• Supplementary file 4. Seroconversion versus MSD cytokine profiling. Results of Spearman correlation analysis between seroconversion indices and cytokines, chemokines, and other immune factors measured by multiplex immunoassays using Meso Scale Discovery (MSD) technology. Column A indicates the MSD analyte name, column B indicates the Spearman *rho* value, column C indicates the p-value, and column D indicates the adjusted p-value using the Benjamini–Hochberg method.

• Supplementary file 5. Seroconversion versus mass cytometry (MC). Results of Spearman correlation analysis between seroconversion indices and immune cell subsets measured by MC. Separate tabs are used for the results obtained from different parent lineages: all live cells, all CD45+ live cells, B cells, T cells, CD4+ T cells, CD8+ T cells, myeloid dendritic cells (mDCs), and monocytes. In each tab, column A indicates the population measured, column B indicates parent lineage, column C indicates the Spearman *rho* value, column D indicates the p-value, and column E indicates the adjusted p-value using the Benjamini–Hochberg method.

• Supplementary file 6. Seroconversion versus immune factors. Results of Spearman correlation analysis between seroconversion indices and circulating immune factors measured by mass spectrometry, SOMAscan assays, or multiplex immunoassays with MSD technology. Column A indicates the analyte name in the platform, column B indicates the name used for display in the corresponding figure, column C indicates the platform used to measure the indicated analyte, column D indicates the Spearman *rho* value, column E indicates the p-value, and column F indicates the adjusted p-value using the Benjamini–Hochberg method.

• Supplementary file 7. IFNA2 versus mass cytometry (MC). Results of Spearman correlation analysis between levels of IFNA2 measured by multiplex immunoassays using MSD technology versus immune cell subsets measured by MC. Column A indicates the immune cell population among all live cells, column B indicates the lineage used to calculate cell frequencies, column C indicates the

Spearman *rho* value, column D indicates the p-value, and column E indicates the adjusted p-value using the Benjamini–Hochberg method.

• Supplementary file 8. Seroconversion versus complement. Results of Spearman correlation analysis between seroconversion indices and components of the complement pathways measured by mass spectrometry or SOMAscan assays. Column A indicates the unique identifier within the platform, column B indicates the name use for display in the corresponding figure, column C indicates the platform used to measure the indicated analyte, column D indicates the Spearman *rho* value, column E indicates the p-value, and column E indicates the adjusted p-value using the Benjamini–Hochberg method.

• Supplementary file 9. Seroconversion versus hemostasis network. Results of Spearman correlation analysis between seroconversion indices and factors involved in control of hemostasis measured by mass spectrometry or SOMAscan assays. Column A indicates the unique identifier within the platform, column B indicates the name used for display in the corresponding figure, column C indicates the platform used to measure the indicated analyte, column D indicates the Spearman *rho* value, column E indicates the p-value, and column E indicates the adjusted p-value using the Benjamini–Hochberg method.

• Supplementary file 10. Seroconversion versus clinical laboratory values. Results of Spearman correlation analysis between seroconversion indices and clinical laboratory values closest to the time of the research blood draw. Column A indicates the clinical laboratory parameter, column B indicates the Spearman *rho* value, column C indicates the p-value, and column D indicates the adjusted p-value using the Benjamini–Hochberg method.

• Supplementary file 11. Mass cytometry (MC) antibody table. List of antibodies used in MC. Column A indicates the antibody target, column B indicates the element conjugated to the antibody, column C indicates the mass of the element, column D indicates the manufacturer, column E indicates the catalog number, column F indicates the clone number, and column G indicates the type of stain protocol used (fixed, live, or fixed with permeabilization).

• Supplementary file 12. Immune cell type definition. List of immune cell subsets defined by mass cytometry. Column A indicates the population identified, column B indicates definition based on gating strategy employed, and column C indicates the parent lineage.

• Transparent reporting form

## Data availability

All data generated for this manuscript is made available through the online researcher gateway of the COVIDome Project, known as the COVIDome Explorer, which can be accessed at http://covidome.org/. Differences between COVID19 negative and positive patients can be visualized in the 'Impact of COVID19' dashboards for each -omics dataset. Differences between sero-low and sero-high COVID19 patients can be visualized in the 'Impact of Seroconversion' dashboards. The mass spectrometry proteomics data have been deposited to the ProteomeXchange Consortium via the PRIDE partner repository (67) with the dataset identifier PXD022817. The mass cytometry data has been deposited in Flow Repository under the link: https://flowrepository.org/id/RvFrSYioKeU-dYHXdkTD9TQPAXt4PqdkB5eie82h11JgAGSCQIneLKpcKd81Nzgwq. The SOMAscan Proteomics, MSD Cytokine Profiles, and Sample Metadata files have been deposited in Mendeley under entry doi:10.17632/2mc6rrc5j3.1. The RNA-seq data have been deposited in NCBI Gene Expression Omnibus, with the accession number GSE167000 (https://www.ncbi.nlm.nih.gov/geo/query/acc.cgi?acc=GSE167000).

The following datasets were generated:

| Author(s) | Year | Dataset title | Dataset URL | Database and Identifier |
|---|---|---|---|---|
| Baxter R | 2020 | COVID CytoF | https://flowrepository.org/id/RvFrSYioKeU-dYHXdkTD9TQ-PAXt4PqdkB5eie82h11J-gAGSCQI- | Flow Repository, FR-FCM-Z367 |

| | | | neLKpcKd81Nzgwq | |
| --- | --- | --- | --- | --- |
| Hansen KC | 2021 | COVIDome Mass Spec Proteomics | https://www.ebi.ac.uk/pride/archive/projects/PXD022817 | PRIDE, PXD022817 |
| Galbraith MD, Espinosa JM | 2020 | COVIDOme datasets | https://dx.doi.org/10.17632/2mc6rrc5j3.1 | Mendeley Data, 10.17632/2mc6rrc5j3.1 |
| Galbraith MD, Espinosa JM | 2021 | PolyA RNA-seq from whole blood of Sars-COV2-negative and -positive subjects | https://www.ncbi.nlm.nih.gov/geo/query/acc.cgi?acc=GSE167000 | NCBI Gene Expression Omnibus, GSE167000 |

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
