## [Decision Letter]

**Acceptance summary:**

In this study, the authors use a systems immunology approach to document innate and adaptive immune responses during clincal SARS-CoV-2 infection. This general impact of this work is a better understanding of COVID19 pathobiology and more specifically, the identification of serum antibodies as a novel classification framework to understand COVID-19 disease course and associated changes.

**Decision letter after peer review:**

Thank you for submitting your article "Seroconversion stages COVID19 into distinct pathophysiological states" for consideration by *eLife*. Your article has been reviewed by three peer reviewers, and the evaluation has been overseen by a Reviewing Editor and Sara Sawyer as the Senior Editor. The following individual involved in review of your submission has agreed to reveal their identity: Bernard Khor (Reviewer #3).

Essential revisions:

Since most of the reviewer comments are seeking clarification to the text and/or current data sets, we paste below all original comments and ask that the authors prepare a detailed response to each.

Reviewer #1 (Recommendations for the authors):

Authors have used tempered statement and recommendations. Their analyses are well done and they limit themselves to associations they discover, while proposing why this may indeed be true. I think this work is well executed and belongs to the current growing COVID19 scientific documents.

Reviewer #2 (Recommendations for the authors):

The study is well designed and written.

Only few comments for the Discussion:

1) Several factors (as C Reactive Protein, Complement factors, IP-10, etc…) are increased in COVID-19 are upregulated also in other infections, as HIV, or TB or others. Please comment, reporting also the results by Santos et al., 2019; Esmail et al., 2018; Lubbers et al., 2020.

2) When the topic of JAK inhibitors is discussed, please, refer to: Kalil et al., 2020; Cantini et al., 2020.

3) For the clinicians that will read the paper: is any possibility, based on the results reported, to draw a figure with the potential clinical algorithm? ex: based on the outcome of serology and or plasmatic IFN and so on?

Reviewer #3 (Recommendations for the authors):

Cohort description: It would be helpful to understand the control cohort a little better. Were these also hospitalized patients? non-ICU? What types of diseases? p values in Supplementary file 1 might be helpful. The d-dimer is interesting – were there some DIC patients. Also normal ranges for clinical values may be helpful.

z-scores: In general it'd be helpful to understand how these were calculated – e.g. from the control data along, or from all the data in aggregate. In the SARS-CoV2 Ab assay, presumably this is calculated from the control data.

Seronegative/seropositive: I think the seroconversion index is creative and useful. However, I am uncomfortable with the terms seronegative/seropositive, which have been defined here as the lower/upper half of Ab titer. This is not the regular connotation of the term, which would be defined, for example based on the upper limit of negative controls. It may be that these thresholds coincide in this study; if so that should be explicitly shown.

How does seroconversion index correlate with duration/severity of disease? Is there a simple canonical relationship with time in this non-severe cohort? If there are relationships different from time that would really emphasize the utility of this approach. Even examining a few canonical differences would be helpful.

In the heat maps, it could be helpful to indicate differences that are statistically significant, for example with an * in the box.

In Figure 2A, how do absolute values change? For example, are Th1/Th17 increased because of specific depletion of other subsets? That would be interesting even as a supplement.

Is there pSTAT or transcriptional evidence of decreased IFN signaling with seroconversion? Or is this more about decreased IFN milieu?

In the hemostasis section, care might be used with the work thrombocytopenia, which has a clinical threshold that does not appear crossed in most subjects. Decreased platelets might be more accurate or I suppose transient relative thrombocytopenia. What is the interpretation of the D-dimer result? In the absence of reference ranges it's hard to know if it's high in seropositive (and control) patients for example due to clot resolution, or low in seronegative patients.

Similarly, hypoalbuminemia has a clinical threshold. Reference ranges would be helpful; it almost appears that all subjects in all cohorts are hypoalbuminemic which can't be the case?

Is decreased hepatic synthesis an alternative hypothesis that needs to be addressed? Factor VIII levels may be helpful in that regard.

---

## [Author Response]

Reviewer #2 *(Recommendations for the authors)*:The study is well designed and written.Only few comments for the Discussion:1) Several factors (as C Reactive Protein, Complement factors, IP-10, etc…) are increased in COVID-19 are upregulated also in other infections, as HIV, or TB or others. Please, comment it reporting also the results by Santos et al., 2019; Esmail et al., 2018; Lubbers et al., 2020.

Thanks for bringing these references to our attention, which are now cited in the revised manuscript.

2) When the topic of JAK inhibitors is discussed, please, refer to: Kalil et al., 2020; Cantini et al., 2020.

Thank you, these references are now cited in the revised manuscript.

3) for the clinicians that will read the paper: is any possibility, based on the results reported, to draw a figure with the potential clinical algorithm? ex: based on the outcome of serology and or plasmatic IFN and so on?

We are very grateful for this comment, which inspired us to investigate the possibility of defining a classifier that could discriminate the sero-low versus sero-high groups by using ratios of clinical laboratory values elevated upon seroconversion with albumin (ALB, which is decreased upon seroconversion) as the denominator. Indeed, the WBC/ALB, D-dimer/ALB, Platelets/ALB, ALP/ALB, ANC/ALB, BUN/ALB, and ALC/ALB ratios were all significantly higher in the sero-high group (new Figure 6F, Figure 6—figure supplement 1D-E). Using cut-off values that would capture >70% of the sero-high group when used individually, we noted that using any 2 of these cut offs concurrently would provide >90% specificity and >75% sensitivity in identifying a sero-high patient (new Figure 6G). Increasing the classifier criteria to pass any 3 or any 4 of these ratios increases specificity at the cost of sensitivity (new Figure 6G). Although the clinical utility of this classifier would require validation efforts in much larger cohorts, it nonetheless illustrates the variable clinical presentation of COVID19 pathology along a quantitative spectrum of seroconversion.

Reviewer #3 *(Recommendations for the authors)*:Cohort description: It would be helpful to understand the control cohort a little better. Were these also hospitalized patients? non-ICU? What types of diseases? p values in Supplementary file 1 might be helpful. The d-dimer is interesting – were there some DIC patients. Also normal ranges for clinical values may be helpful.

Thank you for these comments, which we address in the revised manuscript with a much more comprehensive Supplementary file 1, where we now include: (a) Information on co-morbidities for both the COVID19 negative and positive cohorts, as well as ICU admission at any time after the research blood draw; (b) p values comparing both the control group versus the COVID19 cohort, but also the sero-low versus the sero-high groups within the COVID19 cohort, and (c) Normal ranges of laboratory values and % of patients outside those ranges in each group for key values (e.g. lymphopenia, thrombocytopenia, hypoalbuminemia).

Noteworthy, the IRB protocol overseeing the collection of the biospecimens used in this study imposes a “cell suppression” measure that prevent us from disclosing data on any clinical variable for which the sample size is less than 10 participants. Therefore, many variables in Supplementary file 1 are labeled as “<10”.

Additional information about the control group is also provided in the Materials and methods, explaining that this group was composed of COVID19 negative participants in the same hospital system receiving care for an assortment of diverse comorbidities, none of them in critical condition (i.e. non ICU) at the time of the research blood draw.

Regarding the comment about d-dimer and DIC, we provide information in the Supplementary file 1 about “coagulopathy” as defined by the Elixhauser Comorbidity List. Out of the 71 COVID19 positive patients with complete clinical data, 15 are annotated as having been affected by “coagulopathy”. As appreciated by the reviewer, this sample size would not be sufficient for well powered comparisons, and larger studies would be required to define differences in the -omics datasets between those with and without DIC and/or thrombotic disorders. Nevertheless, prompted by the reviewer’s comment, in the revised Discussion we refer to a recent report describing the results of analysis of ~2M medical records from COVID19 patients in the USA, showing that d-dimer levels increase over the course of hospitalization (and more so in patients with severe COVID19) (1), further supporting the staging model proposed in our manuscript, whereby high d-dimer is a marker of “Stage 2”.

z-scores: In general it'd be helpful to understand how these were calculated – e.g. from the control data along, or from all the data in aggregate. In the SARS-CoV2 Ab assay, presumably this is calculated from the control data.

Thank you for this comment. This had been described in the Materials and methods text: “Z-scores were calculated from the adjusted concentration values for each epitope (or analyte) in each sample, based on the mean and standard deviation of COVID19-negative samples.” To help readers and add clarity, we have added this information to all figures legends where Z scores are used.

Seronegative/seropositive: I think the seroconversion index is creative and useful. However, I am uncomfortable with the terms seronegative/seropositive, which have been defined here as the lower/upper half of Ab titer. This is not the regular connotation of the term, which would be defined, for example based on the upper limit of negative controls. It may be that these thresholds coincide in this study; if so that should be explicitly shown.

We agree with this comment and decided to prevent confusion by using instead the “sero-low” and “sero-high” labels throughout the manuscript. Given that the clinical definition of “seropositivity” may vary depending on the clinical assay used and locality, we agree with the reviewer that adhering to the low and high ranges of the seroconversion index calculated from our cohort would prevent misinterpretation.

How does seroconversion index correlate with duration/severity of disease? Is there a simple canonical relationship with time in this non-severe cohort? If there are relationships different from time that would really emphasize the utility of this approach. Even examining a few canonical differences would be helpful.

This is an important point that could not be fully addressed with our cross-sectional cohort, current sample size, and limited data about timing on “onset of symptoms”, but which can be addressed to some degree via analysis of published data.

As hypothesized by the reviewer, it is expected that the seroconversion index would increase over time in symptomatic patients, eventually plateauing (specially for IgGs as measured in our assays). In the Discussion, we cited reports documenting the timeline of seroconversion in COVID19 relative to onset of symptoms, and we have now expanded this fragment to address how the seroconversion index would behave over time. The revised fragment reads as follows:

“The temporal sequence of seroconversion relative to the onset of COVID19 symptoms has been already established (2, 3). […] In this model, a seroconversion index based on IgG levels would increase over the first two to four weeks after symptom onset, with concomitant changes in COVID19 pathophysiology”.

Regarding the relationship between seroconversion index and disease severity, we proposed in our Discussion that longer time to seroconversion (as expected in the elderly) could lead to increased tissue damage during the “inflammatory” Stage 1 and therefore increased downstream consequences, such as coagulopathy, in Stage 2. In the revised Discussion, we cite new results derived from analysis of ~2M COVID19 cases in the USA via the National COVID19 Cohort Collaborative (N3C) (currently under peer review), which strongly support this overall notion. The revised fragment reads as follows:

“Given the cross-sectional nature of our study, our model should be challenged with longitudinal analysis of the biosignatures reported here. […] Furthermore, this analysis supports the notion that disease severity is associated with increased and more prolonged inflammation in Stage 1 (as assessed by CRP levels) along with higher levels of D-dimer later on (1).”

In the heat maps, it could be helpful to indicate differences that are statistically significant, for example with an * in the box.

Thanks for this suggestion, which we have incorporated in the revised manuscript. In the new figures, an asterisk indicates a significant difference relative to the control COVID19 negative group, and a + symbol indicates a significant difference between the sero-low versus sero-high groups.

In Figure 2A, how do absolute values change? For example, are Th1/Th17 increased because of specific depletion of other subsets? That would be interesting even as a supplement.

We thank the reviewer for this comment. We agree with the importance of assessing absolute cell numbers when possible. For calculation of the absolute cell number one would need (1) a complete blood cell count (CBC with differential) matching the research blood draw (or very close in time), and (2) the volume of the blood sample from which the PBMCs were isolated. Unfortunately, a CBC linked to the research blood draw was not available for all of the study subjects, which would force us to remove many of the samples from the analysis, thus reducing statistical power. Based on the clinical and research data we do have, we could provide the relative cell numbers from each population, using the total number of cells acquired on the mass cytometer as denominator. However, this information would be redundant with the relative frequency that we are currently displaying in the figures. To address this comment, we have revised the text to emphasize to readers that our data is expressed as immune cell frequency within the indicated parent lineages.

Is there pSTAT or transcriptional evidence of decreased IFN signaling with seroconversion? Or is this more about decreased IFN milieu?

Thanks for this inquiry, which we address with revisions to the text and new data. Upon seroconversion, we notice a significant decrease in the actual levels of the several IFN ligands in circulation (with the exception of IFN beta) (e.g. Figure 3B). In turn, this is accompanied by decreased levels of circulating IFN-inducible proteins, such as IP10/CXCL10 (Figure 3C), so the data indicate that the phenomenon is more about decreased IFN milieu, with consequent decrease in ISG transcriptional output. In the revised manuscript, we include new data showing that the mRNA levels for key ISGs, such as IP10/CXCL10, ISG15, MX1, and IFIT1, are indeed significantly increased only in the sero-low group, returning to baseline or near baseline levels in the sero-high group (new Figure 3—figure supplement 1C).

In the hemostasis section, care might be used with the work thrombocytopenia, which has a clinical threshold that does not appear crossed in most subjects. Decreased platelets might be more accurate or I suppose transient relative thrombocytopenia. What is the interpretation of the D-dimer result? In the absence of reference ranges it's hard to know if it's high in seropositive (and control) patients for example due to clot resolution, or low in seronegative patients.

Thanks for these constructive comments, which we address in the revised manuscript with modifications to the text and by providing the reference ranges for both platelets and D-Dimer values (and all other clinical labs) in Supplementary file 1. The platelet count is indeed significantly lower in the sero-low group, but, as noted by the reviewer, only ~30% fall below the commonly used reference range. Therefore, following the reviewer’s guidance, we have revised the text to use “depletion of platelets”, instead of “thrombocytopenia” where appropriate.

Regarding the D-dimer results, the mean value for the sero-low group hovers around the threshold of 500 ng/mL commonly used in the clinic to define a “positive” D-dimer test but the mean value for sero-high group is significantly higher (Supplementary file 1), D-dimer values were determined for only a small fraction of the control group and are highly variable, so there is no significant difference with the sero-low group, so the suggestion by the reviewer to interpret the results in relationship to the reference threshold is much appreciated. The text has been revised accordingly to clarify interpretation of the D-dimer results.

Similarly, hypoalbuminemia has a clinical threshold. Reference ranges would be helpful; it almost appears that all subjects in all cohorts are hypoalbuminemic which can't be the case?Is decreased hepatic synthesis an alternative hypothesis that needs to be addressed? Factor VIII levels may be helpful in that regard.

Thanks for these constructive comments. We are now providing the reference ranges for albumin (3.5-5.6 g/dL) in the new Supplementary File 1. The mean for sero-low patients is within the range (3.6 -/+ 0.4) but below the range for sero-high patients (3.2 -/+ 0.6), this difference is significant (p=0.00082) and it translates into a higher fraction of true hypalbuminemia among sero-high patients (62.5 %). Following the reviewer’s guidance, we use “albumin depletion” versus “hypoalbuminemia” where appropriate. Also, thanks to the reviewer’s comment, we realized that we had omitted the “log2” preface on the Y axis label in Figure 6D and E, which gave the impression that all cohorts were below the reference range, which is not the case. We have fixed this in the new figures.

Regarding decreased hepatic synthesis of albumin and other liver proteins in seroconverted patients, this is a non-mutually exclusive possibility, and lower albumin in blood could be due to both lesser production and/or higher interstitial leakage, which we acknowledge in the manuscript. Many liver-derived proteins are depleted in sero-high patients relative to sero-low patients, not just albumin, including other major plasma proteins and coagulation factors (see Figure 5A and Figure 6—figure supplement 1C). Factor VIII (F8) was not significantly associated with seroconversion (at FDR of 10%, Supplementary file 3), but F7, F2, F11, F12, and F10 were all significantly decreased in the sero-high group, so decreased liver production is certainly a possibility. In the revised manuscript, we emphasize more clearly that more pronounced hypoalbuminemia could be due to decreased liver synthesis and/or stronger interstitial leakage.

References:

1) Bennett TD. The National COVID Cohort Collaborative: Clinical Characterization and Early Severity Prediction. medRxiv. 2021. doi: https://doi.org/10.1101/2021.01.12.21249511.

2) Chen Y, Tong X, Li Y, Gu B, Yan J, Liu Y, Shen H, Huang R, Wu C. A comprehensive, longitudinal analysis of humoral responses specific to four recombinant antigens of SARS-CoV-2 in severe and non-severe COVID-19 patients. PLoS pathogens. 2020;16(9):e1008796. Epub 2020/09/12. doi: 10.1371/journal.ppat.1008796. PubMed PMID: 32913364; PMCID: PMC7482996.

3) Seow J, Graham C, Merrick B, Acors S, Pickering S, Steel KJA, Hemmings O, O'Byrne A, Kouphou N, Galao RP, Betancor G, Wilson HD, Signell AW, Winstone H, Kerridge C, Huettner I, Jimenez-Guardeno JM, Lista MJ, Temperton N, Snell LB, Bisnauthsing K, Moore A, Green A, Martinez L, Stokes B, Honey J, Izquierdo-Barras A, Arbane G, Patel A, Tan MKI, O'Connell L, O'Hara G, MacMahon E, Douthwaite S, Nebbia G, Batra R, Martinez-Nunez R, Shankar-Hari M, Edgeworth JD, Neil SJD, Malim MH, Doores KJ. Longitudinal observation and decline of neutralizing antibody responses in the three months following SARS-CoV-2 infection in humans. Nat Microbiol. 2020. Epub 2020/10/28. doi: 10.1038/s41564-020-00813-8. PubMed PMID: 33106674.